# TMS: Trajectory-Mixed Supervision for On-Policy Self Distillation

**Rana Muhammad Shahroz Khan** [1]   **Zijie Liu** [1]   **Zhen Tan** [2]   **Charles Fleming** [3]   **Tianlong Chen** [1]

## Abstract

Reinforcement Learning (RL) and Supervised Fine-Tuning (SFT) are the two dominant paradigms for enhancing Large Language Model (LLM) performance on downstream tasks. While RL often preserves broader model capabilities (retention) better than SFT, it comes with significant costs: complex reward engineering, instability, and expensive on-policy sampling. In contrast, SFT is efficient but brittle, often suffering from catastrophic forgetting due to **Supervision Mismatch**: the divergence between the model's evolving policy and static training labels. We address this trade-off with **Trajectory-Mixed Supervision (TMS)**, a reward-free framework that uses trajectory-aligned, near-policy supervision harvested from the model's own historical checkpoints. TMS reduces *Policy-Label Divergence (PLD)* within SFT-style training and preserves multiple plausible solution modes, mitigating a key source of forgetting in standard SFT. Experiments across reasoning (MATH, GSM8K) and instruction-following benchmarks demonstrate that TMS effectively shifts the accuracy–retention Pareto frontier. While RL remains the strongest retention baseline, TMS significantly outperforms standard and iterative SFT, narrowing the gap to RL without requiring reward models or verifiers. Mechanistic analysis shows that KL-to-base is the strongest cross-method predictor of forgetting, while PLD provides a complementary diagnostic of supervision mismatch within SFT-style methods.

## 1. Introduction

The post-training of large language models (LLMs) has largely converged to a simple recipe: Supervised Fine-

---
[1]The University of North Carolina at Chapel hill [2]Arizona State University [3]Cisco. Correspondence to: Rana <shahroz@cs.unc.edu>.

*Proceedings of the $43^{rd}$ International Conference on Machine Learning*, Seoul, South Korea. PMLR 306, 2026. Copyright 2026 by the author(s).

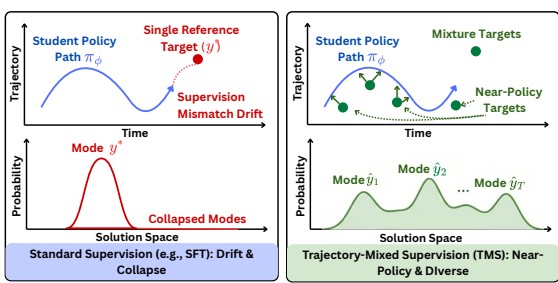

*Figure 1.* Single-reference supervision (e.g., SFT) induces supervision-mismatch drift and mode collapse; trajectory-mixed supervision (TMS) samples near-policy targets across training, preserving diverse solution modes.

Tuning (SFT) on curated demonstrations, followed by preference or RL-based alignment when higher reliability is required (Ouyang et al., 2022; Christiano et al., 2023). SFT remains the default in practice due to its simplicity, stability, and efficiency in teaching task formats and high-quality behaviors. However, mounting evidence suggests that SFT often comes with a severe side effect: capability collapse or catastrophic forgetting (Kotha et al., 2024; Zhang et al., 2025). While improving performance on the target downstream distribution, SFT-tuned models can regress in general reasoning, linguistic diversity, and even pre-existing safety guardrails (Dong et al., 2023). As a result, these models become brittle under out-of-distribution (OOD) prompts, where small shifts in instruction style or reasoning demands can trigger disproportionate failures (Sclar et al., 2024; Perez et al., 2021; Wei et al., 2023).

In contrast, on-policy RL-style post-training, such as PPO (Schulman et al., 2017) and verifier-driven variants like GRPO (Shao et al., 2024), is often observed to better preserve broad generalization while still improving target-task performance (Ouyang et al., 2022; Bai et al., 2022a; Shao et al., 2024). A key distinction is *where supervision comes from*: RL continually induces training targets from the model's current policy, keeping learning coupled to the model's evolving behavior, whereas SFT relies on static labels that do not adapt as the policy shifts (Ross et al., 2011; Schulman et al., 2017). Despite these benefits, RL is often harder to adopt at scale because it typically requires reward modeling or verifiers for credit assignment and incurs substantial on-policy sampling overhead (Casper et al.,

2023). This leaves practitioners with a persistent dilemma: SFT is easy and efficient but prone to over-specialization and forgetting, while RL is more robust but more complex. These observations raise a natural question:

> **Q.** *Can we recover some retention benefits associated with on-policy optimization without reward models, verifiers, or full RL training loops?*

Existing explanations (e.g., on-policy data or KL regularization) (Chen et al., 2025; Chu et al., 2025; Jin et al., 2025a; Shenfeld et al., 2025) offer useful intuition, but they do not directly yield a simple, reward-free algorithm that can be dropped into standard SFT pipelines. We bridge this gap by re-framing what makes SFT fragile.

**Key observation: SFT optimizes a moving policy against fixed targets.** Although SFT is often framed as learning a fixed input–output mapping, the policy $\pi_{\theta_t}$ evolves throughout training while the supervision distribution $q(y|x)$ remains static. This induces *temporal supervision mismatch*: as the model shifts probability mass toward alternative (often valid) trajectories, single-reference cross-entropy can produce large corrective gradients that over-enforce the reference and suppress diversity, especially on tasks with non-unique solutions (Figure 1). In contrast, on-policy RL continually re-couples optimization targets to the current policy, which helps preserve broad capabilities.

**Our approach: measure mismatch and fix it via trajectory supervision.** We quantify mismatch with *Policy-Label Divergence (PLD)* and propose *Trajectory-Mixed Supervision (TMS)*: a reward-free post-training method that samples supervision from historical checkpoints along a single training trajectory. By mixing near-policy targets across time, TMS reduces mismatch drift and preserves historically plausible solution modes, approximating one stabilizing ingredient of on-policy optimization without reward models.

**In Summary**, we contribute the following:

> ❶ **KL/PLD framing.** We use KL-to-base as the primary cross-method retention diagnostic, and formalize *Policy-Label Divergence (PLD)* as a complementary measure of supervision mismatch within SFT-style training.
>
> ❷ **Reward-free, near-policy post-training.** We propose *Trajectory-Mixed Supervision (TMS)*, which mixes supervision targets from historical checkpoints to reduce PLD and provide trajectory-aligned supervision without reward models, verifiers, or RL loops.
>
> ❸ **Empirical and mechanistic validation.** Across model families and scales, TMS improves the accuracy–retention trade-off on reasoning and instruction-

following benchmarks, and KL/PLD-based drift metrics explain why TMS shifts the Pareto frontier.

## 2. Related Work

Extended related works are listed in Appendix B.

**Post-Training Dynamics and Forgetting.** Post-training like SFT and RL has become a standard paradigm for adapting foundation models to downstream tasks, often yielding substantial gains in instruction-following, reasoning, and task-specific utility (Ouyang et al., 2022; DeepSeek-AI et al., 2024; Lu et al., 2025; Kirk et al., 2024; Luo et al., 2022; Xie et al., 2024). However, such post-training optimization can induce capability drift and catastrophic forgetting, especially under distribution shift or continual updates (Luo et al., 2025; Li et al., 2024b; Kotha et al., 2024). While prior work mitigates forgetting by constraining parameters or replaying data, these approaches typically assume a fixed supervision signal during post-training; in contrast, we study forgetting as a consequence of training a changing model under static supervision.

**Preference Optimization and On-Policy Alignment.** Recent work on post-training alignment spans both on-policy reinforcement learning and offline preference optimization objectives (Rafailov et al., 2023; Bhatia et al., 2025; Schulman et al., 2017; Annadani et al., 2025; Yan et al., 2025). A group of studies finds that RL-based post-training can exhibit substantially less forgetting and better generalization than supervised fine-tuning (SFT) under distribution shift (Chen et al., 2025; Chu et al., 2025; Lai et al., 2026; Jin et al., 2025b). Our work shows that one source of this retention advantage lies in keeping supervision close to the model's evolving support, and demonstrates that this effect can be approximated in a reward-free manner.

**Trajectory Learning and Data Selection Curricula.** Recent work increasingly leverages model-generated trajectories as training signals, enabling self-distillation and iterative refinement without additional labels (Wang et al., 2023; Zelikman et al., 2022; Guo et al., 2025; Yuan et al., 2025; Bai et al., 2022b). Empirical studies suggest that training on distributions induced by the evolving policy can substantially reduce forgetting relative to static supervision (Chu et al., 2025; Lai et al., 2026; Jin et al., 2025a). Unlike prior trajectory-based methods focused on performance or curricula, our work uses trajectories to align supervision with an evolving policy and reduce forgetting without rewards.

## 3. Preliminaries

### 3.1. Post-Training Formulation

We consider post-training of a language model parameterized by $\theta$, initialized from a pre-trained base policy $\pi_{\theta_0}$. The

model maps inputs $x \in \mathcal{X}$ to outputs $y \in \mathcal{Y}$.

**Supervised Fine-Tuning (SFT).** SFT assumes a static dataset $\mathcal{D} = \{(x_i, y_i^*)\}_{i=1}^{N}$ drawn from a reference distribution $q(y|x)$ (e.g., human experts, verifiers, or a teacher model). The objective minimizes the negative log-likelihood of the reference labels under the current policy $\pi_\theta$ (with $\theta$ updated over steps $t$):

$$\mathcal{L}_{\text{SFT}}(\theta) = -\mathbb{E}_{(x,y^*)\sim\hat{p}_\mathcal{D}}\left[\log \pi_\theta(y^*|x)\right]. \quad (1)$$

Crucially, in SFT the target distribution $q$ remains fixed throughout training, while the policy $\pi_\theta$ evolves, inducing a temporal mismatch analyzed in subsequent sections.

**Reinforcement Learning (RL).** RL fine-tuning optimizes a reward signal $r(x, y)$ over model outputs $y$ conditioned on prompts $x$. A common formulation maximizes expected reward while regularizing deviation from the base model via a KL penalty:

$$\max_\theta \ \mathbb{E}_{x\sim\mathcal{D}_{\text{prompt}}}\left[J(\theta; x)\right],$$

$$J(\theta; x) = \mathbb{E}_{y\sim\pi_\theta(\cdot|x)}[r(x, y)] - \beta\, D_{\text{KL}}\big(\pi_\theta(\cdot|x) \,\|\, \pi_{\theta_0}(\cdot|x)\big).$$

Here, $\mathcal{D}_{\text{prompt}}$ denotes the prompt distribution (which may overlap with inputs in $\mathcal{D}$). RL is *on-policy* with respect to $(y|x)$: training trajectories (responses) are sampled from the evolving policy itself, so the response distribution shifts with model capabilities.

### 3.2. Evaluation Framework: Target vs. Retention

We distinguish improvement on the trained task from preservation of existing capabilities.

❶ **Target Performance** ($S_{\text{tgt}}$): performance on downstream tasks $\mathcal{T}_{\text{tgt}}$ used for post-training.

❷ **Retention Performance** ($S_{\text{ret}}$): performance on a held-out suite $\mathcal{T}_{\text{ret}}$ not seen during post-training, measuring retained capabilities and robustness to distribution shift.

### 3.3. The Forgetting Metric

Let $S_b(\pi)$ be the score of policy $\pi$ on benchmark $b \in \mathcal{T}_{\text{ret}}$, and $\Delta_b = S_b(\pi_{\theta_t}) - S_b(\pi_{\theta_0})$. We define the Forgetting Score as average degradation across the retention suite, considering only negative changes:

$$\mathcal{F}(\pi_{\theta_t}) = \frac{1}{|\mathcal{T}_{\text{ret}}|} \sum_{b\in\mathcal{T}_{\text{ret}}} \min(\Delta_b, 0). \quad (2)$$

In this formulation, $\mathcal{F} \le 0$; more negative values indicate worse catastrophic forgetting.

## 4. What Exactly Fails in SFT?

Prior work (Chen et al., 2025; Chu et al., 2025; Jin et al., 2025a; Shenfeld et al., 2025) often attributes the superior

*Table 1.* **Failure Mode A: Mismatch drift during SFT.**

| Step | Train NLL↓ | Val NLL↓ | MATH Acc↑ | ARC-C Acc↑ |
|------|-----------|----------|-----------|------------|
| 0 (Base) | 1.85 | 1.88 | 48.1% | 66.9% |
| 500 | 0.85 | 0.92 | 52.5% | 59.1% |
| 1000 | 0.42 | 0.44 | 56.5% | 52.3% |
| **2000** | **0.15** | **0.67** | **61.2%** | **49.5%** |
| 5000 | 0.04 | 0.98 | 62.4% | 45.2% |

retention of RL fine-tuning to broad factors such as KL regularization or the use of preference/negative samples. We argue that forgetting under SFT is not monolithic. Instead, it arises from two largely independent failure modes that emerge when optimizing an evolving policy against a static supervision distribution: *(A) temporal supervision mismatch* and *(B) mode collapse under single-reference cross-entropy*.

### 4.1. Failure Mode A: Temporal Supervision Mismatch

The core structural property of SFT is that the supervision distribution $q(y|x)$ is fixed, while the policy $\pi_{\theta_t}$ evolves over optimization steps.

❶ **Mechanism.** As training progresses, $\pi_{\theta_t}$ often places mass on responses that are semantically correct but differ from the single reference $y^*$ in surface form (e.g., phrasing, reasoning style, intermediate steps). When $\pi_{\theta_t}(y^*|x)$ becomes small, token-level cross-entropy induces large gradients that pull the policy back toward $y^*$, producing non-local updates that can interfere with features supporting unrelated capabilities. Intuitively, the optimizer is forced to spend capacity matching a rigid template rather than preserving general representations.

❷ **Evidence and observable signatures.** We observe a characteristic *knee point*: training NLL decreases monotonically, while a held-out estimate of the same objective on $\mathcal{D}_{\text{val}}$ (from the same task distribution) improves early and then degrades at later checkpoints. Table 1 illustrates the pattern on QWEN2.5-1.5B (Yang et al., 2024) fine-tuned with SFT on MATH (Hendrycks et al., 2021c). While training NLL continues to fall, held-out NLL reaches a minimum at step 1000 and then increases. In the same interval, retention on ARC-C (Clark et al., 2018) collapses, even though target MATH accuracy continues to improve slightly. This decoupling indicates that SFT can "converge" on the training objective while drifting away from generalizable behavior.

**Hypothesis A (H-A).** *A major driver of SFT forgetting is supervision mismatch drift: as the policy evolves away from the fixed supervision distribution, held-out PLD (proxied by validation NLL) can rise after the initial fit phase and mark brittle reference matching. Forgetting may begin earlier, but a rising held-out PLD tail diagnoses worsening supervision mismatch within SFT-style training.*

*Table 2.* **Failure Mode B: Mode collapse under SFT.**

| Method | Pass@1↑ | Pass@100↑ | SC-Acc↑ | AnsEnt↑ |
|---|---|---|---|---|
| Base Model | 48.1% | 62.0% | 56.0% | **2.45** |
| Standard SFT | 62.4% | 66.0% | 68.0% | 0.85 |
| RL (GRPO) | **63.5%** | 73.9% | **75.2%** | 1.35 |
| TMS (Ours) | 62.8% | **74.0%** | 70.9% | 1.28 |

## 4.2. Failure Mode B: Mode Collapse Under Single-Reference CE Targets

Standard SFT applies token-level cross-entropy to a single reference trajectory $y^*$. For many reasoning and code tasks, the solution set is non-unique: there exist many valid outputs $y'$ that reach the correct answer.

❶ **Mechanism.** By concentrating likelihood mass on $y^*$, SFT implicitly suppresses alternative valid trajectories $y'$, encouraging collapse onto a narrow mode. This is particularly harmful when robustness requires maintaining support over multiple valid reasoning paths (*e.g.*, paraphrases).

❷ **Evidence and diagnostics.** We quantify mode collapse using coverage and diversity proxies computed from $K$ sampled rollouts per prompt under fixed decoding across methods: (i) *Pass@K* (coverage); (ii) *Self-consistency accuracy* (SC-Acc; majority-vote accuracy over $K$ samples); and (iii) *Answer entropy* (AnsEnt; entropy of the empirical answer distribution; higher implies broader support). On MATH, SFT improves Pass@1 but sharply reduces answer entropy, consistent with collapsing to a narrow solution mode (Table 2). In contrast, on-policy RL (GRPO) and our TMS method preserve substantially higher entropy while maintaining strong coverage (Pass@100). Throughout, we compute Pass@K/SC-Acc/AnsEnt with the same sampling protocol for all methods, so the observed differences reflect changes in model support rather than evaluation artifacts.

> **Hypothesis B (H-B).** *On tasks with non-unique solutions, single-reference SFT induces mode collapse, which harms robustness and contributes to downstream degradation. Methods that preserve multi-modal support improve coverage and retention.*

### 4.3. Policy-Label Divergence (PLD)

To diagnose the supervision side of these failures, we introduce **Policy-Label Divergence (PLD)**. While KL-to-base measures how far a post-trained policy has moved from the base model ($\pi_{\theta_0}$), PLD measures the tension between the model's current *beliefs* ($\pi_{\theta_t}$) and the *supervision* distribution ($q$) it is trained to match. Thus, KL-to-base is our primary cross-method retention diagnostic, while PLD is a mechanism-level diagnostic for SFT-style methods with an explicit supervision distribution.

**Formal definition (forward form aligned with SFT).** We

define PLD as the expected forward KL (equivalently cross-entropy up to a constant) between the supervision distribution and the policy:

$$\text{PLD}(\theta_t; q) = \mathbb{E}_{x \sim \mathcal{D}_x} \left[ D_{\text{KL}}(q(\cdot|x) \,\|\, \pi_{\theta_t}(\cdot|x)) \right]. \quad (3)$$

In the ideal case, $q(y|x)$ would represent the full ground-truth posterior. In standard single-reference SFT, $q$ is typically a Dirac delta $\delta_{y^*}$ concentrated on a single reference solution $y^*$. In this setting, minimizing PLD is equivalent to minimizing the usual token-level cross-entropy objective.

**The generalization gap (PLD drift).** The key distinction arises when we estimate PLD on *held-out* inputs drawn from the same task distribution rather than the training samples themselves. Let $\mathcal{D}_{\text{train}}$ be the training set and $\mathcal{D}_{\text{val}}$ a held-out set from the same distribution:

❏ **Memorization phase:** On $\mathcal{D}_{\text{train}}$, PLD decreases monotonically as the model fits the reference trajectories.

❏ **Mismatch drift phase:** On $\mathcal{D}_{\text{val}}$, PLD may reach a minimum early in training and then increase. We interpret this rising tail as **supervision mismatch drift**: the policy is being pulled toward increasingly brittle reference-matching rather than maintaining generalizable representations.

For single-reference supervision, we approximate held-out PLD using validation negative log-likelihood (NLL):

$$\widehat{\text{PLD}}_{\text{val}}(\theta_t) = -\frac{1}{|\mathcal{D}_{\text{val}}|} \sum_{(x,y^*) \in \mathcal{D}_{\text{val}}} \log \pi_{\theta_t}(y^*|x). \quad (4)$$

A model can maintain high target accuracy (e.g., correct final answers) while $\widehat{\text{PLD}}_{\text{val}}$ increases, indicating growing disagreement with the supervision template. PLD is not intended as a cross-method retention predictor across all post-training algorithms; for RL methods, which do not optimize a single SFT-style label distribution $q$, KL-to-base remains the more comparable global metric.

## 5. Method: Trajectory-Mixed Supervision

We introduce **Trajectory-Mixed Supervision (TMS)**, a reward-free post-training procedure that addresses both failure modes in Section 4. TMS replaces static single-reference supervision with a *trajectory mixture* of the model's own intermediate policies, turning the optimization path into a curriculum. TMS is not online on-policy RL: the student trains on a fixed buffer harvested from historical checkpoints. Its intended role is narrower, providing trajectory-aligned, near-policy targets that (i) reduce supervision mismatch drift and (ii) broaden support over valid solution modes that mitigate mode collapse. We formalize this perspective and relate TMS to mismatch and mode preservation in Appendix D.

## 5.1. Core Algorithm

**Stage 1: Trajectory harvesting.** Given a training set $\mathcal{D} = \{(x_i, y_i^*)\}_{i=1}^N$, we run a standard post-training procedure for a fixed budget and record $T$ intermediate checkpoints $\{\theta_1, \ldots, \theta_T\}$ at uniformly spaced steps.[1] For each checkpoint $t$ and input $x$, we generate a response $\hat{y}^{(t)}(x) \sim \pi_{\theta_t}(\cdot|x)$ and store the mapping $(x, t) \mapsto \hat{y}^{(t)}(x)$. We denote the resulting trajectory buffer by

$$\mathcal{H} = \{\hat{y}^{(t)}(x) \; : \; x \in \mathcal{D}_x, \; t \in \{1, \ldots, T\}\},$$

where $\mathcal{D}_x$ denotes the empirical distribution over inputs $\mathcal{D}$.

**Stage 2: Trajectory-mixed supervision distribution.** For each input $x$, TMS defines an empirical mixture supervision distribution supported on the trajectory outputs:

$$m(\cdot|x) = \frac{1}{T} \sum_{t=1}^{T} \delta_{\hat{y}^{(t)}(x)}(\cdot). \tag{5}$$

Optionally, we interpolate with the original reference distribution $q(\cdot|x)$ (e.g., $\delta_{y^*}$ for single-reference SFT) using a mixing weight $\alpha \in [0, 1]$:

$$q_\alpha(\cdot|x) = \alpha \, q(\cdot|x) + (1 - \alpha) \, m(\cdot|x). \tag{6}$$

In the main setting, we use the trajectory mixture ($\alpha = 0.25$) unless stated otherwise.

**Stage 3: Student training.** We train a student policy $\pi_\phi$ from the same initialization as the baseline (e.g., $\phi_0 = \theta_0$) by minimizing the forward KL (equivalently cross-entropy) to the mixture supervision:

$$\min_\phi \quad \mathbb{E}_{x \sim \mathcal{D}_x} \Big[ D_{\mathrm{KL}}\big(q_\alpha(\cdot|x) \,\|\, \pi_\phi(\cdot|x)\big)\Big]. \tag{7}$$

Operationally, this corresponds to sampling $t \sim \mathrm{Uniform}\{1, \ldots, T\}$ (or sampling from $q$ with probability $\alpha$) and applying standard token-level NLL on it. The full algorithm can be found in Appendix C.

# 6. Theoretical Analysis

This section provides a formal lens for the empirical failure modes in Section 4. We first clarify why single-reference SFT can be structurally misaligned with robustness on non-unique tasks. We then show that changes in expected task performance are controlled by distributional drift from the base model, yielding a principled justification for KL-to-base as a predictor of forgetting.

## 6.1. Motivation: Why static single-reference supervision can be misaligned

SFT optimizes a forward-KL (cross-entropy) projection onto a fixed supervision distribution $q(y|x)$. When $q$ is

---

[1] In all experiments, we use $T = 10$ unless otherwise stated.

*sparse*, e.g., a Dirac delta $\delta_{y^*}$ representing a single reference trajectory, the projection can encourage concentration of probability mass on that trajectory, suppressing alternative valid solutions. This aligns with Failure Mode B (Section 4) and motivates relaxing $q$ into a broader supervision distribution that preserves multi-modal support.

**Proposition 6.1** (**Single-reference projection encourages concentration**). *For a fixed input $x$, minimizing the cross-entropy $\mathbb{E}_{y \sim q(\cdot|x)}[-\log \pi_\theta(y|x)]$ with $q(\cdot|x) = \delta_{y^*}$ is equivalent to maximizing $\log \pi_\theta(y^*|x)$. This objective provides no positive training signal for alternative valid outputs $y' \neq y^*$, and therefore can reduce support over the set of valid solutions when capacity is limited or optimization is continued beyond the point of generalization.*

Proposition 6.1 is not a claim that SFT must always collapse modes, but that its *training signal* is inherently one-sided under single-reference supervision. TMS replaces the fixed $q$ by a trajectory mixture whose support includes multiple historically plausible outputs, directly addressing this.

## 6.2. Bounding behavior change via distribution drift

We now show that distributional drift from the base model upper-bounds changes in expected performance on held-out tasks. This is a generic drift bound rather than a TMS-specific guarantee, and it formalizes why KL-to-base is a meaningful predictor of forgetting. It complements PLD, which diagnoses mismatch with supervision.

❒ **Assumption 1 (Bounded score).** For a given held-out task, let $f_x : \mathcal{Y} \to [0, 1]$ be a bounded scoring function for prompt $x$ (e.g., indicator of correctness, normalized reward).

❒ **Assumption 2 (Prompt distribution).** Prompts are drawn from a fixed evaluation distribution $x \sim \mathcal{D}_{\mathrm{eval}}$ (e.g., the held-out suite prompts).

**Theorem 6.2** (**Forgetting is controlled by KL drift**). *Let $\pi_0(\cdot|x)$ be the base policy and $\pi_\theta(\cdot|x)$ a fine-tuned policy. Then the change in expected score on $\mathcal{D}_{\mathrm{eval}}$ is bounded by the KL divergence from the base:*

$$\left| \mathbb{E}_{x \sim \mathcal{D}_{\mathrm{eval}}}\big[ \mathbb{E}_{y \sim \pi_\theta(\cdot|x)} f_x(y)\big] - \mathbb{E}_{x \sim \mathcal{D}_{\mathrm{eval}}}\big[ \mathbb{E}_{y \sim \pi_0(\cdot|x)} f_x(y)\big]\right|$$
$$\leq \mathbb{E}_{x \sim \mathcal{D}_{\mathrm{eval}}}\Big[\sqrt{2 \, D_{\mathrm{KL}}(\pi_\theta(\cdot|x) \,\|\, \pi_0(\cdot|x))}\Big].$$

**Proof sketch.** Fix a prompt $x$. The difference in expected score between $\pi_\theta$ and $\pi_0$ can be written as an inner product between the score function $f_x$ (bounded in $[0, 1]$) and the signed measure $(\pi_\theta - \pi_0)$. By $L_1/L_\infty$ duality, this difference is at most twice the total variation distance between $\pi_\theta(\cdot|x)$ and $\pi_0(\cdot|x)$. Pinsker's inequality upper-bounds total variation by $\sqrt{\frac{1}{2}\mathrm{KL}}$, yielding the stated bound. The full proof is in Appendix E.

*Table 3.* **Main results across models.** Target-task performance $S_{\text{tgt}}$ (higher is better), cross-task transfer to other target tasks $S_{\text{xfer}}$ (Avg $\Delta$, lower is better), and held-out retention. Parentheses denote change vs. the base model; **best** / second-best are highlighted.

| Method | Target Tasks $S_{\text{tgt}}$ ↑ | | | | | | Cross-task $S_{\text{xfer}}$ ↓ | Held-out Retention ↑ | | |
|---|---|---|---|---|---|---|---|---|---|---|
| | Math500 | MATH | GSM8K | Count. | IFEval | MMLU | Avg $\Delta$ other tgt | ARC-C | Hotpot | SafetyBench |
| *Qwen-2.5-1.5B-Instruct* | | | | | | | | | | |
| Base | 46.6 | 48.1 | 68.9 | 13.2 | 38.3 | 30.8 | – | 66.9 | 43.1 | 35.1 |
| Standard SFT | 58.4 | 62.4 | 77.3 | 25.2 | 54.1 | **39.0** | ↓ 26.2 | 42.3(↓ 24.6) | 19.4(↓ 23.7) | 31.4(↓ 3.7) |
| Self-SFT | 54.9 | 57.1 | 73.1 | 29.4 | 47.3 | 34.7 | ↓ 18.4 | 51.4(↓ 15.5) | 27.5(↓ 15.6) | 34.3(↓ 0.8) |
| Final-SFT | 57.1 | 61.9 | **78.0** | 25.1 | 53.1 | 36.8 | ↓ 24.3 | 48.3(↓ 18.6) | 25.3(↓ 17.8) | 34.0(↓ 1.1) |
| REINFORCE | 56.9 | 60.1 | 75.2 | 26.1 | 49.3 | 34.7 | ↓ 3.4 | 63.8(↓ 3.1) | 40.1(↓ 3.0) | 35.0(↓ 0.1) |
| GRPO | 58.9 | **63.5** | 76.3 | **35.2** | 53.6 | 35.2 | ↓ 1.2 | **66.7**(↓ 0.2) | **42.0**(↓ 1.1) | **36.3**(↑ 1.3) |
| TMS (Ours) | **59.0** | 62.8 | 76.8 | 32.4 | **54.4** | 38.6 | ↓ 2.9 | 65.4(↓ 1.4) | 41.4(↓ 1.7) | 34.8(↓ 0.3) |
| *Qwen-2.5-3B-Instruct* | | | | | | | | | | |
| Base | 60.2 | 62.0 | 85.2 | 29.6 | 60.6 | 43.8 | – | 80.5 | 53.6 | 35.7 |
| Standard SFT | 76.4 | 80.3 | **90.2** | 49.2 | 71.3 | **54.0** | ↓ 39.2 | 42.7(↓ 37.8) | 24.1(↓ 29.5) | 31.1(↓ 4.6) |
| Self-SFT | 74.2 | 77.4 | 87.3 | 48.6 | 67.9 | 47.2 | ↓ 29.3 | 51.8(↓ 28.7) | 33.1(↓ 20.5) | 33.6(↓ 2.1) |
| Final-SFT | 75.8 | 79.6 | 89.5 | 48.1 | 70.2 | 51.4 | ↓ 35.1 | 45.2(↓ 35.3) | 28.4(↓ 25.2) | 33.2(↓ 2.5) |
| REINFORCE | 75.1 | 78.2 | 88.4 | 47.3 | 67.2 | 49.1 | ↓ 4.1 | 78.6(↓ 1.9) | 51.4(↓ 2.2) | **35.5**(↓ 0.2) |
| GRPO | 77.4 | **81.5** | 90.1 | **54.2** | 70.9 | 50.8 | ↓ 1.9 | **80.2**(↓ 0.3) | **53.1**(↓ 0.5) | 35.4(↓ 0.3) |
| TMS (Ours) | **77.8** | 81.1 | 88.9 | 51.3 | **72.0** | 53.6 | ↓ 2.3 | 79.2(↓ 1.3) | 51.7(↓ 1.9) | 35.0(↓ 0.7) |
| *LLaMA-3.1-8B-Instruct* | | | | | | | | | | |
| Base | 47.2 | 46.0 | 84.1 | 18.6 | 67.0 | 46.2 | – | 84.1 | 52.6 | 33.8 |
| Standard SFT | 66.7 | 64.6 | **89.6** | 39.2 | 75.2 | **55.1** | ↓ 32.9 | 53.2(↓ 30.9) | 28.2(↓ 24.4) | 30.1(↓ 3.7) |
| Self-SFT | 60.4 | 60.4 | 87.4 | 36.4 | 71.6 | 50.8 | ↓ 24.9 | 64.2(↓ 19.9) | 39.1(↓ 13.5) | 32.6(↓ 1.2) |
| Final-SFT | 64.3 | 63.1 | 88.8 | 37.4 | 73.8 | 53.2 | ↓ 31.4 | 60.8(↓ 23.3) | 36.2(↓ 16.4) | 32.2(↓ 1.6) |
| REINFORCE | 63.9 | 61.8 | 88.1 | 34.8 | 72.2 | 49.3 | ↓ 4.9 | 82.1(↓ 2.0) | 50.6(↓ 2.0) | 33.6(↓ 0.2) |
| GRPO | **70.5** | **65.4** | 89.4 | **41.2** | 75.6 | 50.8 | ↓ 0.9 | **83.8**(↓ 0.3) | **52.1**(↓ 0.5) | **34.1**(↑ 0.3) |
| TMS (Ours) | 66.3 | 64.9 | 89.2 | 40.4 | **76.1** | 54.6 | ↓ 2.3 | 83.1(↓ 1.0) | 51.4(↓ 1.2) | 33.0(↓ 0.8) |

# 7. Experiments

## 7.1. Experimental Setup

**Task groups and evaluation protocol.** We evaluate post-training along three axes: **(1) In-domain target performance** on the task used for training, **(2) cross-task transfer** to other downstream tasks not seen during training, and **(3) held-out retention** on capability benchmarks never used for post-training. For details, please refer to Appendix F.

**Datasets.** We consider the following task groups:

**(1) Target tasks (used for training).** We fine-tune on **Math-500** (Lightman et al., 2023), **MATH** (Hendrycks et al., 2021c), **GSM8K** (Cobbe et al., 2021), and **Countdown** (Pan et al., 2025) (verifiable multi-step reasoning), and also on **IFEval** (Zhou et al., 2023) (instruction following) and **MMLU** (Hendrycks et al., 2021a;b) (knowledge). These tasks span both non-unique solution spaces (math) and more rigid evaluation formats (IFEval/MMLU), enabling targeted tests of our hypotheses. For MATH-500, MATH, GSM8K, and Countdown, the downstream performance is measured using the test split of the datasets. For IfEval, we evaluate on the training set, while for MMLU, the downstream performance is measured on MMLU-Pro (Wang et al., 2024).

**(2) Cross-task transfer (held-out downstream tasks).** For

each run trained on a specific target task $\mathcal{T}_{\text{train}}$ (e.g., MATH), we additionally evaluate the post-trained model on the remaining target tasks $\mathcal{T}_{\text{other}} = \mathcal{T}_{\text{tgt}} \setminus \{\mathcal{T}_{\text{train}}\}$. We report the *average delta* relative to the base model on $\mathcal{T}_{\text{other}}$ as a cross-task generalization score, capturing whether improvements reflect transferable capability or brittle specialization.

**(3) Held-out retention suite (never trained on).** To measure catastrophic forgetting and robustness beyond the trained distribution, we evaluate on **ARC-Challenge** (Clark et al., 2018) (commonsense/science), **HotpotQA** (Yang et al., 2018) (multi-hop reading), and safety benchmarks **WildGuardTest** (Han et al., 2024), **SafetyBench** (Zhang et al., 2023), **WildJailbreak** (Jiang et al., 2024).

**Models.** We evaluate across instruction-tuned open-weight LLMs: *LLaMA-3.1-8B-Instruct* (Grattafiori et al., 2024), *Qwen-2.5-7B-Instruct* (Yang et al., 2024), *LLaMA-3.2-1B-Instruct*, *Qwen-2.5-1.5B-Instruct*, and *Qwen-2.5-3B-Instruct*. Appendix G.3 reports an additional 70B pilot. For all SFT-based models, we use *LLaMA-3.3-70B* as the teacher to generate our CoTs and filter them based on corrections. For RL algorithms, we use a simple binary verifier based on answer correctness.

**Baselines.** We compare against strong post-training baselines: **(1) Standard SFT** (ground-truth/teacher references),

**(2) Self-SFT** (single-snapshot synthetic supervision from $\pi_{\theta_0}$ outputs), **(3) Final-SFT** (single-snapshot synthetic supervision from converged SFT outputs), and on-policy RL baselines **(5) GRPO** (Shao et al., 2024) and **(6) REINFORCE**. For all fine-tuning, we fine-tune the full model. Appendix G.1 additionally reports Rephrase-SFT and $T$-matched single-snapshot controls.

**Metrics reported.** For each training run on $\mathcal{T}_{\text{train}}$, we report: **(i) Target score** $S_{\text{tgt}}(\mathcal{T}_{\text{train}})$, **(ii) Cross-task transfer** $S_{\text{xfer}} = \frac{1}{|\mathcal{T}_{\text{other}}|} \sum_{b \in \mathcal{T}_{\text{other}}} \left( S_b(\pi_{\text{post}}) - S_b(\pi_{\theta_0}) \right)$, and **(iii) Retention/forgetting** on $\mathcal{T}_{\text{ret}}$, reported both as absolute scores and the aggregate forgetting score $\mathcal{F}$ (defined in Section 3).

## 7.2. Main Results

We now answer RQ1–RQ4 using Table 3. Full results can be found in Appendix H. Across model families and scales, a consistent pattern emerges: **standard SFT reliably improves target-task accuracy but induces large cross-task interference and severe retention collapse**, while **TMS achieves near-RL retention with SFT-level performance**.

> **RQ1 (Target performance).** *Does TMS outperform standard SFT on the trained (in-domain) target task under matched compute?*

**Target tasks: TMS preserves (and often improves) the SFT gains.** Across all three models, TMS matches or exceeds SFT on most target benchmarks, indicating that trajectory mixing does not "wash out" learning signal. For *Qwen-2.5-1.5B*, TMS attains the best Math500 score and the best IFEval score, while remaining competitive on the remaining targets (Table 3). For *Qwen-2.5-3B*, TMS improves over SFT on Math500 and IFEval, and stays close on MATH/GSM8K/MMLU. For *LLaMA-3.1-8B*, TMS improves IFEval and MMLU, while matching SFT on the remaining targets. Overall, TMS consistently retains the primary benefit of post-training, *i.e.*, in-domain competence, while avoiding the instability.

> **RQ2 (Forgetting).** *Does TMS reduce catastrophic forgetting on a held-out retention suite relative to standard SFT?*

**Retention: TMS sharply reduces the collapse induced by SFT.** The most striking effect in Table 3 is that SFT causes pronounced degradation on held-out capabilities, despite improving target-task accuracy. For *Qwen-2.5-1.5B*, standard SFT severely degrades ARC-C and HotpotQA (*e.g.*, ARC-C drops from 66.9 to 42.3), whereas TMS retains near-base performance. The same qualitative pattern appears at larger scales: on *Qwen-2.5-3B*, SFT collapses ARC-C from

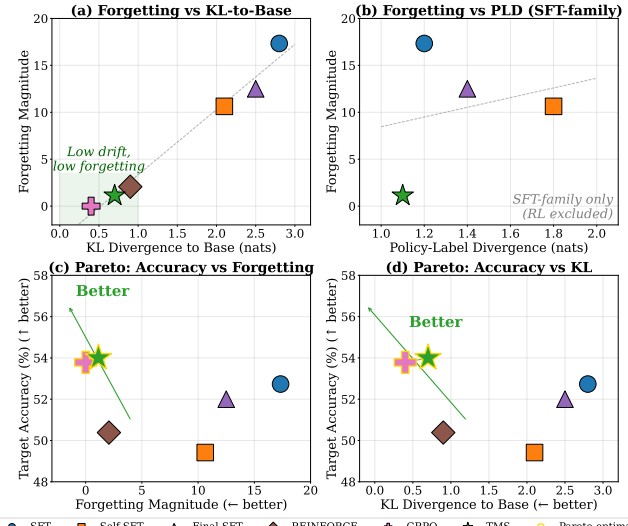

*Figure 2.* **RQ5 (Mechanistic law + Pareto).** (a) Forgetting magnitude increases with KL-to-base. (b) PLD aligns with forgetting within the SFT-family (RL excluded). (c–d) Pareto frontiers show TMS moving closer to the accuracy–retention frontier relative to SFT and single-snapshot baselines.

80.5 to 42.7, while TMS preserves 79.2; on *LLaMA-3.1-8B*, SFT drops ARC-C from 84.1 to 53.2, while TMS retains 83.1. These large retention gaps demonstrate that trajectory mixing is an effective intervention against catastrophic forgetting, consistent with the mismatch and mode-collapse diagnoses in Section 4.

> **RQ3 (RL parity without rewards).** *Can TMS match the accuracy–retention tradeoff of on-policy RL baselines* without *reward models or verifiers?*

**Accuracy–retention frontier: TMS tracks on-policy RL while remaining reward-free.** On-policy RL baselines (GRPO/REINFORCE) exhibit a characteristic signature across models: strong retention with low cross-task drift, but at non-trivial pipeline complexity. TMS attains a similar operating point without rewards, critics, or online on-policy optimization. For *Qwen-2.5-3B*, TMS is close to GRPO on retention (ARC-C: 79.2 vs. 80.2; HotpotQA: 51.7 vs. 53.1) while remaining competitive on target tasks (e.g., Math500: 77.8 vs. 77.4; IFEval: 72.0 vs. 70.9). For *LLaMA-3.1-8B*, TMS similarly tracks GRPO on retention, with comparable target accuracy. The gap is most visible in the cross-task transfer column: GRPO tends to minimize cross-task drop, while TMS incurs a modestly larger drop, yet still dramatically improves over SFT. Taken together, these results place TMS near the same Pareto region as on-policy RL, supporting the claim that *near-policy supervision* can be approximated through trajectory mixing alone.

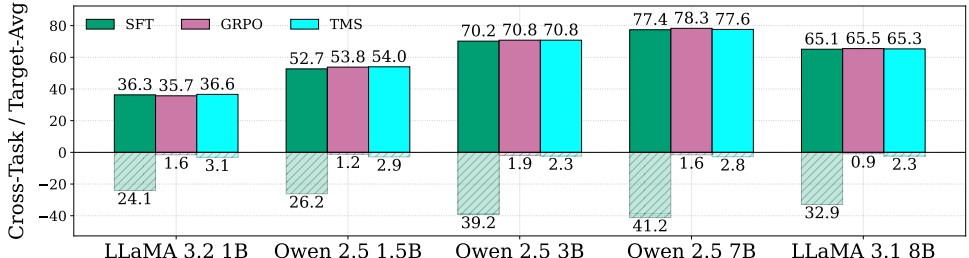

*Figure 3.* **RQ6 (Scaling):** Target Avg (higher is better) and Cross-task (lower is better) across model scales. TMS tracks GRPO-style low cross-task drift while matching SFT-level target accuracy.

**RQ4 (Beyond single-snapshot supervision).** *Are TMS gains explained by a single synthetic-supervision checkpoint or does the* trajectory mixture *provide additional benefit?*

**Beyond single-checkpoint supervision: trajectory mixing is essential.** Self-SFT and Final-SFT are single-snapshot synthetic-supervision baselines: they replace oracle labels with outputs from one policy snapshot, removing part of the supervision mismatch while keeping the training procedure identical. Across all models, these baselines improve retention relative to standard SFT but remain consistently below TMS. For *Qwen-2.5-1.5B*, Self-SFT improves ARC-C to 51.4, yet TMS achieves 65.4; similar gaps hold on HotpotQA. For *Qwen-2.5-3B*, Self-SFT recovers ARC-C to 51.8, while TMS preserves 79.2; and for *LLaMA-3.1-8B*, Self-SFT achieves 64.2 on ARC-C versus 83.1 for TMS. Appendix G.1 further shows that TMS outperforms Rephrase-SFT and $T$-matched single-snapshot controls, indicating that **mixing supervision across the training trajectory**, rather than selecting any single checkpoint or rewriting labels once, is the key ingredient.

**RQ5 (Task dependence).** *When does TMS help most: which task properties (e.g., non-unique solutions vs. rigid formats) predict larger gains?*

**Task dependence: larger gains appear on flexible, multi-solution targets.** Table 3 suggests a consistent pattern aligned with our dissection in Section 4: TMS yields its clearest advantages on targets where solution spaces are non-unique and supervision mismatch is common (*e.g.*, math and instruction-following). On such tasks, TMS typically matches SFT on target metrics while substantially improving held-out retention. In contrast, on more rigid or multiple-choice-style targets (*e.g.*, MMLU), TMS is generally competitive, but the gap to SFT is larger.

### 7.3. Mechanistic and Secondary Analyses

For further analyses, please refer to Appendix G.

**RQ6 (Mechanistic law).** *When we compute (i) KL-to-base on new-task prompts, (ii) PLD to the supervision distribution of each method, and (iii) forgetting on the held-out suite, do KL/PLD reliably track forgetting, and does TMS shift the Pareto frontier relative to SFT?*

**KL-to-base predicts forgetting, and TMS occupies the low-drift/low-forgetting regime.** Figure 2a shows a strong monotonic relationship between KL-to-base and forgetting magnitude: methods that drift farther from the base model tend to forget more. In particular, SFT-family baselines exhibit both high KL drift and large forgetting, while GRPO and TMS cluster in the lower-left "low drift, low forgetting" region. Across SFT/TMS/RL methods, KL-to-base has Spearman correlation 0.93 with forgetting, making it the strongest global predictor in our analysis.

**PLD is most informative within the SFT family.** When PLD is computed w.r.t. the SFT-style supervision distribution, it is not directly comparable across RL methods. Accordingly, Figure 2b reports PLD–forgetting trends within the SFT-family only (RL excluded). In this regime, higher PLD aligns with higher forgetting, with Spearman correlation 0.76 over SFT/TMS methods and 0.84 after excluding the early-fit phase. Thus PLD is best viewed as a complementary diagnostic of supervision mismatch within SFT-style training, not as a replacement for KL-to-base.

**Pareto analysis: TMS shifts the accuracy–retention frontier relative to SFT.** Figure 2c–d visualize the core tradeoffs. In the accuracy vs. forgetting plane (c), and accuracy vs. KL plane (d), SFT and single-snapshot baselines are dominated: they incur substantially higher forgetting and drift at comparable accuracy. In contrast, TMS lies on (or very near) the Pareto-optimal set alongside GRPO, indicating that it achieves a better accuracy–retention tradeoff than SFT.

**RQ7 (Scaling).** *How do TMS effects vary with model scale, in both target performance and retention?*

**Scaling: TMS remains stable across 1B–8B and preserves the GRPO-like low cross-task drift.** Figure 3 summarizes Target Avg and Cross-task transfer across five

model sizes. Two trends are consistent: (i) Target Avg increases predictably with scale for all methods, and TMS matches SFT and GRPO closely; (ii) cross-task drift under SFT remains large across scales, while TMS stays close to GRPO with small cross-task drops. This indicates the TMS advantage is not a small-model artifact and persists across families and sizes. A 70B pilot in Appendix G.3 shows the same direction with smaller absolute retention gaps, suggesting that larger models are more robust but still benefit from trajectory mixing.

## 8. Conclusion

We studied why post-training with static supervision (SFT) often improves target-task performance at the cost of broad capability retention. We introduced **Trajectory-Mixed Supervision (TMS)**, a reward-free alternative that turns the SFT optimization path into a supervision curriculum, reducing *policy–label mismatch* and mitigating mode collapse. Across model families and scales, TMS matches SFT-level target gains while approaching on-policy RL (GRPO/REINFORCE) in retention and cross-task stability. Mechanistically, KL-to-base is the strongest cross-method predictor of forgetting, while PLD diagnoses supervision mismatch within SFT-style training. These results suggest that *on-policy-like stability can be recovered without rewards* by designing supervision to track the model's evolving support, making TMS a simple drop-in primitive for robust post-training.

## Impact Statement

This paper proposes *Trajectory-Mixed Supervision (TMS)*, a reward-free post-training method intended to improve the accuracy–retention tradeoff of large language models by reducing supervision mismatch and catastrophic forgetting. If adopted, TMS could yield practical benefits such as more reliable continual updates, lower dependence on reward models or complex RL pipelines, and improved stability of safety-related behaviors under post-training.

At the same time, improving retention and robustness can also increase the capability of models in ways that may be misused (e.g., retaining stronger general problem-solving ability while being adapted for a specialized domain). In addition, TMS relies on model-generated supervision from intermediate checkpoints; if the base model exhibits harmful or biased behaviors, mixing trajectory outputs may propagate such patterns unless mitigated through careful data curation and safety evaluation. To reduce these risks, we report held-out safety benchmarks (e.g., SafetyBench and jailbreak ASR) and encourage practitioners to pair TMS with standard deployment safeguards, auditing, and domain-appropriate oversight.

Overall, this work aims to advance the science of post-training dynamics and the design of safer, more stable adaptation procedures.

## Acknowledgment

This research was, in part, funded by the CISCO Faculty Award. The views and conclusions contained in this document are those of the authors and should not be interpreted as representing official policies, either expressed or implied, of the funding organizations.

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

# Appendix Contents

## A. LLM Usage Disclosure

To enhance clarity and readability, we utilized LLMs (specifically OpenAI GPT-5) exclusively as a language polishing tool. Its role was confined to proofreading, grammatical correction, and stylistic refinement: functions analogous to those provided by traditional grammar checkers and dictionaries. This tool did not contribute to the generation of new scientific content or ideas, and its usage is consistent with standard practices for manuscript preparation.

## B. Extended Related Work

**Post-Training Dynamics and Forgetting (Extended).**   Prior work has analyzed how neural optimization trajectories reshape the loss landscape and contribute to forgetting in sequential training regimes (Goodfellow et al., 2015; McCloskey & Cohen, 1989). To mitigate drift and forgetting, existing approaches broadly fall into several categories. Regularization-based methods constrain parameter updates to preserve previously learned behaviors, e.g., by penalizing deviations from important weights or reference representations, trading off plasticity for stability (Kirkpatrick et al., 2017; Zenke et al., 2017; Aljundi et al., 2018; Li & Hoiem, 2017; Li et al., 2024a). Replay and rehearsal strategies maintain a small memory of past data (or synthetic samples) and interleave them during continued post-training to prevent collapse on earlier distributions (Lopez-Paz & Ranzato, 2022; Rebuffi et al., 2017; Krawczyk & Gepperth, 2024; Klasson et al., 2022). Parameter-efficient adaptation limits the effective degrees of freedom of post-training, reducing interference with pre-trained knowledge while still enabling task adaptation (Houlsby et al., 2019; Li & Liang, 2021; Hu et al., 2021; Liu et al., 2022; Hu et al., 2023). Together, these lines of work suggest that controlling which data drives post-training and how strongly updates reshape the model are both critical for maintaining capability retention under continual optimization.

**Preference Optimization and On-Policy Alignment (Extended).**   Follow-up work further studies continual post-training and reports that reinforcement fine-tuning can mitigate forgetting over time, while RL fine-tuning can also alleviate OOD forgetting introduced by SFT and improve robustness under distribution shift (Lai et al., 2026; Jin et al., 2025b). Beyond RL and SFT, recent analyses emphasize distributional considerations in reasoning and alignment, showing that matching the training data distribution to a model's own generation distribution can be an important driver of stable improvements (Chandra et al., 2025; Cheng et al., 2025). While these findings highlight the benefits of on-policy training, existing approaches rely on explicit rewards, verifiers, or preference signals.

**Trajectory Learning and Data Selection Curricula (Extended).**   Recent work increasingly leverages model-generated trajectories including rollouts (Wang et al., 2023), intermediate reasoning traces (Zelikman et al., 2022; Guo et al., 2025), and self-evaluations as training signals (Yuan et al., 2025; Bai et al., 2022b), enabling self-distillation (Mukherjee et al., 2023; Agarwal et al., 2024) and iterative refinement without additional labels (Madaan et al., 2023; Zelikman et al., 2024). Beyond RL, analyses of reasoning post-training emphasize distributional considerations, showing that matching the training distribution to a model's own generation distribution can be a key driver of stable improvements, sometimes even more influential than strict correctness of supervision (Chandra et al., 2025; Cheng et al., 2025). These findings naturally connect trajectory learning to dynamic data selection and curriculum learning, where trajectories provide feedback about the learner's evolving state and motivate adaptive sampling strategies that improve robustness and sample efficiency over static training sets (Cheng et al., 2025; Chu et al., 2025).

## C. TMS Algorithm

We provide the full Trajectory-Mixed Supervision (TMS) procedure for transparency and reproducibility. TMS turns a *single* baseline post-training trajectory into a *distribution* over supervision targets. Concretely, instead of training a student against a fixed reference $y^*$ (which can induce supervision-mismatch drift and mode collapse), we harvest intermediate policy outputs along the training path and sample targets from their mixture, yielding near-policy supervision that better preserves diverse solution modes (Figure 1).

**Inputs and outputs.** TMS takes as input a base policy $\pi_{\theta_0}$ and a supervised dataset $\mathcal{D} = \{(x_i, y_i^*)\}_{i=1}^N$. It outputs a post-trained student policy $\pi_\phi$ trained on *trajectory-mixed* targets. The hyperparameter $T$ controls how many checkpoints are used (trajectory resolution), and $\alpha \in [0, 1]$ controls how often we fall back to oracle/teacher labels (quality anchoring).

**Two-stage procedure.** TMS is implemented in two stages. **Stage 1** harvests a trajectory buffer by running a baseline post-training procedure (typically SFT, but any deterministic update path can be used) and saving $T$ checkpoints $\{\theta_1, \ldots, \theta_T\}$.

At each checkpoint $t$, we generate one (or a small number of) sampled outputs $\hat{y}^{(t)}(x)$ for each prompt $x$ (or a representative subset) and store them in a buffer $\mathcal{H}[t, x]$. **Stage 2** trains a student initialized from $\theta_0$ using standard token-level NLL, but with targets drawn from a mixture: with probability $\alpha$ we train on the reference label $y^*$; otherwise we sample a checkpoint index $t \sim p(t)$ (uniform in Algorithm 1 unless otherwise specified) and train on the corresponding trajectory target $\mathcal{H}[t, x]$.

**Practical notes.** (i) The checkpoint distribution $p(t)$ can be uniform (default) or non-uniform (e.g., early-/mid-/late-heavy) to test "window" effects; (ii) storing text sequences is sufficient for our experiments and is cheaper than storing logits; and (iii) to control compute, trajectory harvesting can be performed on a subset of prompts without changing the training objective.

---

**Algorithm 1** Trajectory-Mixed Supervision

---

1: **Input:** Base policy $\pi_{\theta_0}$; dataset $\mathcal{D} = \{(x_i, y_i^*)\}_{i=1}^{N}$; checkpoints $T$; mixing weight $\alpha$.
2: **Stage 1: Harvest trajectory outputs.**
3: Run baseline post-training (e.g., SFT) and save checkpoints $\{\theta_1, \ldots, \theta_T\}$.
4: **for** $t = 1$ to $T$ **do**
5:     **for** each $x \in \mathcal{D}_x$ (or subset) **do**
6:         Generate $\hat{y}^{(t)}(x) \sim \pi_{\theta_t}(\cdot \mid x)$; store in $\mathcal{H}[t, x]$.
7:     **end for**
8: **end for**
9: **Stage 2: Train student on trajectory-mixed supervision.**
10: Initialize student $\phi \leftarrow \theta_0$.
11: **while** training budget not exhausted **do**
12:     Sample minibatch $x$ from $\mathcal{D}_x$.
13:     **for** each $x$ in minibatch **do**
14:         With probability $\alpha$, set $\tilde{y} \leftarrow y^*$.
15:         Else sample $t \sim \text{Uniform}(\{1, \ldots, T\})$ and set $\tilde{y} \leftarrow \mathcal{H}[t, x]$.
16:     Add $(x, \tilde{y})$ to $B_{\text{mix}}$.
17:     **end for**
18:     Update $\phi$ by minimizing token-level NLL on $B_{\text{mix}}$.
19: **end while**
20: **Output:** Student policy $\pi_\phi$.

---

# D. A Unified View Through Mismatch and Modes

Our analysis in Section 4 suggests that post-training instability is not a single phenomenon. Rather, it arises from two orthogonal mechanisms: (i) *supervision mismatch drift* when a moving policy is trained against static supervision, and (ii) *mode collapse* when single-reference cross-entropy suppresses alternative valid solutions. This section provides a unified interpretation of TMS as an intervention that addresses both mechanisms using a single principle: **replace rigid supervision with near-policy, trajectory-supported supervision**.

**(A) Reducing supervision mismatch drift via near-policy targets.** Failure Mode A arises because SFT optimizes $\pi_{\theta_t}$ against a fixed supervision distribution $q(\cdot|x)$; as $\pi_{\theta_t}$ evolves, the mismatch between the current policy and the static targets can grow, which manifests as rising held-out PLD and increasingly corrective gradients. TMS replaces the static target distribution with a *trajectory mixture* $m(\cdot|x)$ whose support is formed by samples from intermediate policies along the same optimization path. Since each component policy $\pi_{\theta_t}$ is itself a feasible model distribution, targets drawn from $m$ are *reachable* throughout training. Empirically, this reduces held-out PLD and mitigates the late-stage mismatch drift observed in Table 1.

**(B) Preventing mode collapse by preserving trajectory support.** Failure Mode B occurs when single-reference cross-entropy concentrates probability mass on one trajectory $y^*$ and implicitly penalizes other valid trajectories. In contrast, the mixture $m(\cdot|x)$ aggregates outputs from checkpoints spanning early (higher-entropy, diverse) to late (more accurate) policies. Training on this mixture encourages the student to assign non-trivial probability to multiple historically plausible solution modes, improving coverage and diversity on non-unique tasks. Consistent with this view, TMS maintains higher Pass@K and answer entropy than single-reference SFT (Table 2), acting as a *mode-preserving regularizer* derived from the model's own learning dynamics.

**(C) Relation to RLFT without rewards.** On-policy RLFT is often more stable because training targets are generated from the current policy (on-policy in $y|x$), keeping updates near-policy. TMS is not online on-policy RL in this strict sense: its student trains on a fixed historical buffer. Instead, the trajectory mixture $m$ can be interpreted as a time-averaged, trajectory-aligned supervision distribution, where targets are sampled from policies that the model actually inhabited during training. This provides a reward-free route to near-policy learning and helps explain why TMS improves the accuracy–retention tradeoff relative to static-label SFT.

# E. Proof of Theorem 6.2

**Assumptions.**

- **Assumption 1 (Bounded scoring function).** For each prompt $x$, the held-out task defines a scoring function $f_x : \mathcal{Y} \to [0, 1]$ (e.g., correctness indicator, normalized reward).

- **Assumption 2 (Evaluation prompt distribution).** Prompts are drawn from a fixed distribution $x \sim \mathcal{D}_{\text{eval}}$ (e.g., prompts from the retention suite).

**Theorem 6.2 (Forgetting is controlled by KL drift).** Let $\pi_0(\cdot|x)$ be the base policy and $\pi_\theta(\cdot|x)$ a fine-tuned policy. Then the change in expected score on $\mathcal{D}_{\text{eval}}$ satisfies

$$\left| \mathbb{E}_{x \sim \mathcal{D}_{\text{eval}}}\left[ \mathbb{E}_{y \sim \pi_\theta(\cdot|x)} f_x(y) \right] - \mathbb{E}_{x \sim \mathcal{D}_{\text{eval}}}\left[ \mathbb{E}_{y \sim \pi_0(\cdot|x)} f_x(y) \right] \right| \leq \mathbb{E}_{x \sim \mathcal{D}_{\text{eval}}}\left[ \sqrt{2\, D_{\text{KL}}(\pi_\theta(\cdot|x) \,\|\, \pi_0(\cdot|x))} \right].$$

**Proof.** Fix an arbitrary prompt $x$. Define the expected score under each policy:

$$\mu_\theta(x) := \mathbb{E}_{y \sim \pi_\theta(\cdot|x)}[f_x(y)], \qquad \mu_0(x) := \mathbb{E}_{y \sim \pi_0(\cdot|x)}[f_x(y)].$$

We bound $|\mu_\theta(x) - \mu_0(x)|$ in three steps.

**Step 1: Expand the difference as an inner product.**

$$\mu_\theta(x) - \mu_0(x) = \sum_{y \in \mathcal{Y}} f_x(y)\big(\pi_\theta(y|x) - \pi_0(y|x)\big).$$

**Step 2: Apply $L_1/L_\infty$ duality (Hölder).** Taking absolute values and using $0 \leq f_x(y) \leq 1$ (Assumption 1),

$$|\mu_\theta(x) - \mu_0(x)| \leq \sum_{y \in \mathcal{Y}} |f_x(y)|\, |\pi_\theta(y|x) - \pi_0(y|x)| \leq \sum_{y \in \mathcal{Y}} |\pi_\theta(y|x) - \pi_0(y|x)|.$$

Recall total variation distance:

$$d_{\text{TV}}(P, Q) := \frac{1}{2} \sum_{y \in \mathcal{Y}} |P(y) - Q(y)|.$$

Therefore,

$$|\mu_\theta(x) - \mu_0(x)| \leq 2\, d_{\text{TV}}(\pi_\theta(\cdot|x),\, \pi_0(\cdot|x)).$$

**Step 3: Convert TV to KL via Pinsker's inequality.** Pinsker's inequality states

$$d_{\text{TV}}(P, Q) \leq \sqrt{\frac{1}{2} D_{\text{KL}}(P\|Q)}.$$

Applying this with $P = \pi_\theta(\cdot|x)$ and $Q = \pi_0(\cdot|x)$ yields

$$|\mu_\theta(x) - \mu_0(x)| \leq 2\sqrt{\frac{1}{2} D_{\text{KL}}(\pi_\theta(\cdot|x) \,\|\, \pi_0(\cdot|x))} = \sqrt{2\, D_{\text{KL}}(\pi_\theta(\cdot|x) \,\|\, \pi_0(\cdot|x))}.$$

**Step 4: Take expectation over prompts.** Taking expectation over $x \sim \mathcal{D}_{\text{eval}}$ (Assumption 2) and using linearity of expectation,

$$\left| \mathbb{E}_x[\mu_\theta(x)] - \mathbb{E}_x[\mu_0(x)] \right| \leq \mathbb{E}_x\left[ \left| \mu_\theta(x) - \mu_0(x) \right| \right] \leq \mathbb{E}_{x \sim \mathcal{D}_{\text{eval}}}\left[ \sqrt{2\, D_{\text{KL}}(\pi_\theta(\cdot|x) \,\|\, \pi_0(\cdot|x))} \right].$$

This completes the proof. ∎

**Corollary E.1.** *Under the assumptions of Theorem 6.2,*

$$\left| \mathbb{E}_x[\mu_\theta(x)] - \mathbb{E}_x[\mu_0(x)] \right| \leq \sqrt{2\, \mathbb{E}_{x \sim \mathcal{D}_{\text{eval}}}[D_{\text{KL}}(\pi_\theta(\cdot|x) \,\|\, \pi_0(\cdot|x))]}.$$

**Proof.** Apply Jensen's inequality to $\sqrt{\cdot}$, which is concave. ∎

# F. Experimental Setup Details

This appendix provides dataset-level details, baseline implementation specifics, and training/inference hyperparameters for reproducibility. Definitions of the core evaluation axes (target, cross-task transfer, retention/forgetting) and aggregate metrics are given in the main paper (§7 and §3); here we describe how they are implemented for each benchmark.

## F.1. Dataset and Benchmark Details

**Target tasks (used for training).**

- **GSM8K.** (Cobbe et al., 2021) 7.5K train / 1K test grade-school math word problems. We fine-tune on the official train split and evaluate on the official test split.

- **MATH.** (Hendrycks et al., 2021c) Competition-level math problems. We fine-tune on the official training split and evaluate on the official test split.

- **Math-500.** (Lightman et al., 2023) A 500-problem subset used for fast evaluation and tuning (reported as a target task when used for training).

- **Countdown.** (Pan et al., 2025) Synthetic arithmetic task (reach target number with given numbers/operators). We generate 10K training examples and a disjoint held-out set using a fixed generator seed.

- **IFEval.** (Zhou et al., 2023) 541 prompts with verifiable instruction constraints; we report strict constraint satisfaction using the official checker.

- **MMLU.** (Hendrycks et al., 2021a;b) 57 subjects; we report standard evaluation accuracy and use an auxiliary training split when fine-tuning on MMLU.

- **ToolAlpaca.** (Tang et al., 2023) A tool-use instruction-following dataset where models must invoke external tools (e.g., calculator/search) and produce multi-step tool-augmented responses. We fine-tune on the provided training split and evaluate on the standard held-out prompts using the ToolAlpaca evaluation protocol.

- **TextVQA.** (Singh et al., 2019) A multimodal visual question answering benchmark focused on reading and reasoning over text in images. We fine-tune on the official training split and evaluate on the standard test split using exact-match accuracy.

- **DocVQA.** (Mathew et al., 2021) A document visual question answering benchmark requiring understanding of scanned documents (layout + OCR text). We fine-tune on the official validation split and evaluate on the standard test split using exact-match accuracy.

**Held-out retention suite (never trained on).**

- **ARC-Challenge.** (Clark et al., 2018) Commonsense/science multiple-choice benchmark.

- **HotpotQA.** (Yang et al., 2018) Multi-hop question answering; we use the standard evaluation setting (distraction).

- **WildGuardTest / SafetyBench / WildJailbreak.** (Han et al., 2024; Zhang et al., 2023; Jiang et al., 2024) Safety and jailbreak robustness benchmarks; we report refusal/violation rates using benchmark-provided evaluators or a fixed judge model (specified in Section F.5).

*Table 4.* **Hyperparameters for full fine-tuning.**

| Model | LR | BS/dev | Grad Acc. | Eff. BS |
|---|---|---|---|---|
| LLaMA-3.2-1B | $1 \times 10^{-4}$ | 4 | 8 | 256 |
| Qwen-2.5-1.5B | $1 \times 10^{-4}$ | 4 | 8 | 256 |
| Qwen-2.5-3B | $2 \times 10^{-5}$ | 2 | 16 | 256 |
| Qwen-2.5-7B | $5 \times 10^{-6}$ | 1 | 32 | 256 |
| LLaMA-3.1-8B | $5 \times 10^{-6}$ | 1 | 32 | 256 |

*Table 5.* **End-to-end cost accounting** (Qwen-2.5-3B / MATH, normalized to SFT).

| Method | FLOPs | Wall-clock | Samples / prompt | Peak storage |
|---|---|---|---|---|
| SFT | $1.0\times$ | $1.0\times$ | 0 | $1.0\times$ |
| TMS | $2.2\times$ | $2.0\times$ | 10 | $1.45\times$ |
| REINFORCE | $3.4\times$ | $3.7\times$ | online | $1.25\times$ |
| GRPO | $3.8\times$ | $4.1\times$ | online group | $1.32\times$ |

### F.2. Baseline Implementations

**Standard SFT.** We minimize token-level NLL on reference completions using the same optimizer, schedule, and training budget as TMS.

**Single-snapshot synthetic-supervision baselines.** We generate one completion per prompt with a fixed decoding policy (temperature/top-$p$; Section F.3), then fine-tune on the resulting synthetic dataset using standard NLL. Self-SFT uses outputs from $\pi_{\theta_0}$; Final-SFT uses outputs from the converged SFT model. We also evaluate $T$-matched versions that generate the same number of synthetic targets per prompt as TMS, using repeated samples from a single snapshot.

**Rephrase-SFT.** To test whether one-shot label-policy matching explains the gains, Rephrase-SFT conditions the base model on the original reference $y^*$ and asks it to produce a behavior-aligned rephrasing. The resulting single rewritten target is then used for standard NLL training.

**GRPO / REINFORCE.** For verifiable tasks (math and Countdown), we use outcome rewards $r(x, y) \in \{0, 1\}$ based on final-answer correctness (exact match or symbolic equivalence). For others, we use LLama-3.3-70B to give a numerical score. GRPO uses group-normalized advantages without a value network; REINFORCE uses a moving-average baseline. All RL runs use the same prompt distribution as SFT/TMS and are matched for total training budget.

### F.3. Decoding and Sampling Policies

**Trajectory harvesting (TMS).** For each checkpoint $t$ and input $x$, we generate $\hat{y}^{(t)}(x) \sim \pi_{\theta_t}(\cdot|x)$ using a fixed sampling policy (temperature/top-$p$), max token limit, and stop rules. Unless otherwise stated, we store one sampled output per $(x, t)$.

**Pass@K and diversity proxies.** To compute Pass@K, majority-vote self-consistency accuracy (SC-Acc), and answer entropy, we draw $K$ independent samples per prompt under the same decoding policy for all methods. We report $K$ and decoding hyperparameters alongside results to ensure comparability. This metric is used in Section 4.

### F.4. Training Hyperparameters and Compute

We use AdamW with a cosine learning rate schedule (3% warmup) and BF16 precision. We train all methods for 2 epochs. Table 4 lists hyperparameters for full fine-tuning.

**Compute environment.** All experiments run on $8\times$ NVIDIA A100 (80GB) GPUs. We use `accelerate` for distributed training and `vLLM` for high-throughput inference and trajectory harvesting.

**End-to-end overhead.** TMS requires an initial trajectory run, trajectory generation, and student training. Table 5 reports normalized end-to-end cost for a representative Qwen-2.5-3B / MATH run. TMS is more expensive than standard SFT, but remains below RL baselines because it does not require online student rollouts, reward modeling, or advantage estimation.

*Table 6.* **Inference tensor parallelism (TP) settings.**

| Model | TP |
|---|---|
| LLaMA-3.2-1B | 8 |
| Qwen-2.5-1.5B | 4 |
| Qwen-2.5-3B | 8 |
| Qwen-2.5-7B | 4 |
| LLaMA-3.1-8B | 8 |

*Table 7.* **Label-matching and snapshot controls** (Qwen-2.5-3B / MATH). Cross-task is Avg $\Delta$ on other target tasks, lower is better.

| Method | MATH↑ | Cross-task↓ | ARC-C↑ | HotpotQA↑ |
|---|---|---|---|---|
| Standard SFT | 80.3 | 39.2 | 42.7 | 24.1 |
| Rephrase-SFT | 79.4 | 17.8 | 60.6 | 38.7 |
| Base-Snapshot SFT ($T$-matched) | 78.3 | 14.7 | 64.7 | 42.8 |
| Final-Snapshot SFT ($T$-matched) | 80.0 | 18.9 | 61.5 | 39.6 |
| TMS | **81.1** | **2.3** | **79.2** | **51.7** |

## F.5. Metric Implementation Details

**Answer extraction and verification (math).** For GSM8K/MATH/Math-500/Countdown, we extract a final answer using a deterministic parser (e.g., `\boxed{}` if present; otherwise the last numeric expression) and verify correctness via exact match or symbolic equivalence using `math-verify`[2]. For Countdown, we additionally verify that the expression uses only permitted numbers/operators.

**IFEval.** We report strict prompt-level constraint satisfaction using the official IFEval checker.

**Helpfulness and safety.** For WildGuardTest/WildJailbreak we report refusal/violation rates using benchmark-provided evaluators. For SafetyBench, we calculate the Accuracy.

**KL-to-base and PLD proxies.** KL-to-base is computed as an expectation over evaluation prompts of the KL between $\pi_\theta(\cdot|x)$ and $\pi_{\theta_0}(\cdot|x)$ using token log-probabilities. Held-out PLD is computed as validation NLL on a held-out split of the training task, as described in Section 4.

## G. Additional Experiments

We present targeted ablations to probe *which aspects of trajectory mixing matter* and whether TMS extends beyond standard single-turn text benchmarks.

### G.1. Label-Matching and Snapshot Controls

**Trajectory mixing is stronger than one-shot label matching.** Table 7 adds controls on Qwen-2.5-3B / MATH to test whether the TMS gain comes only from better label-policy matching or from additional synthetic supervision. Rephrase-SFT improves retention relative to standard SFT, confirming that label-policy matching matters. However, TMS remains substantially stronger than Rephrase-SFT and $T$-matched single-snapshot controls, indicating that mixing across the training trajectory provides value beyond rewriting labels once or sampling more data from a single checkpoint.

### G.2. Mixing Weight Sensitivity

**The default $\alpha = 0.25$ balances oracle anchoring and trajectory diversity.** Table 8 sweeps the oracle-label mixing weight on Qwen-2.5-3B / MATH. Too little oracle anchoring slightly hurts target quality, while too much collapses toward standard SFT and sharply increases cross-task interference. The effect on learning speed is modest; $\alpha$ mainly controls the quality-diversity trade-off.

---

[2]Math-verify: https://github.com/huggingface/Math-Verify

*Table 8.* **Mixing weight sensitivity** (Qwen-2.5-3B / MATH).

| $\alpha$ | MATH↑ | Cross-task↓ | ARC-C↑ | HotpotQA↑ | Steps to 95%↓ |
|---|---|---|---|---|---|
| 0.0 | 80.6 | 4.9 | 76.8 | 49.8 | 1.10× |
| 0.1 | 80.9 | 3.5 | 78.4 | 50.9 | 1.03× |
| 0.25 | **81.1** | **2.3** | **79.2** | **51.7** | 1.00× |
| 0.5 | 80.7 | 4.2 | 77.6 | 50.1 | 0.98× |
| 1.0 (oracle-only SFT) | 80.3 | 39.2 | 42.7 | 24.1 | 0.95× |

*Table 9.* **RQ8 (Alignment & Safety Retention).** Held-out safety evaluation. We report **SafetyBench** (↑) and attack success rates (ASR; ↓) on **WildGuardTest** and **WildJailBreak**. Best/second-best are computed *per model* for each metric.

| Model | Method | SafetyBench↑ | WildGuard ASR↓ | WildJailBreak ASR↓ |
|---|---|---|---|---|
| | Standard SFT | 21.8 | 38.6 | 52.4 |
| | GRPO | **25.0** | **32.0** | **41.4** |
| | TMS (Ours) | 24.6 | 32.3 | 42.2 |
| | Standard SFT | 31.1 | 61.3 | 89.4 |
| | GRPO | **35.4** | 54.8 | 83.7 |
| | TMS (Ours) | 35.0 | **54.2** | **82.9** |
| | Standard SFT | 34.1 | 43.8 | 74.2 |
| | GRPO | **38.6** | **36.6** | **65.2** |
| | TMS (Ours) | 37.1 | 37.1 | 65.9 |

> **RQ8 (Alignment and safety retention).** *Does TMS better preserve helpfulness and safety behavior than SFT on held-out alignment/safety benchmarks?*

**Setup (held-out safety retention).** To isolate safety *retention* rather than safety *learning*, we evaluate all post-trained models on safety benchmarks that are **never used for post-training**. We report (i) **SafetyBench** (↑), a safety QA benchmark scored by accuracy-style criteria, and (ii) jailbreak **attack success rates** (ASR; ↓) on **WildGuardTest** and **WildJailBreak**. Lower ASR indicates stronger refusal/robustness to adversarial prompts. We compare Standard SFT, GRPO, and **TMS** under matched training budgets; best/second-best are computed per model and metric (Table 9).

**Safety retention: SFT erodes safety, while TMS recovers most of the loss.** Across all three models, standard SFT yields noticeably weaker safety behavior, increasing ASR and reducing SafetyBench relative to the stronger baselines. In contrast, TMS consistently reduces ASR and improves SafetyBench compared to SFT, closely tracking the on-policy RL baseline. For example, on **Qwen-2.5-3B**, TMS slightly improves over GRPO on both ASR metrics (WildGuard: 54.2 vs. 54.8; WildJailBreak: 82.9 vs. 83.7) while remaining within a small margin on SafetyBench (35.0 vs. 35.4). On **Qwen-2.5-7B**, TMS again stays close to GRPO (WildJailBreak ASR: 65.9 vs. 65.2), and on **LLaMA-3.2-1B** it recovers a substantial fraction of SFT-induced degradation (e.g., WildJailBreak ASR: 42.2 vs. 52.4 for SFT).

**Interpretation.** These results support the view that safety regressions under continued post-training are a form of *capability drift*: static-label SFT can over-specialize and inadvertently erode previously acquired guardrails. By keeping supervision closer to the model's evolving support, TMS reduces destructive updates and thereby preserves safety behavior that is otherwise fragile under standard SFT.

> **RQ9 (Window vs uniform).** *Which parts of the training trajectory matter most: does concentrating TMS on an intermediate "window" outperform uniform sampling for retention at matched target accuracy?*

**Setup (sampling distributions over checkpoints).** We instantiate TMS with $T=10$ equally-spaced checkpoints along the baseline post-training trajectory (SFT on the same task), saved at fixed step intervals from the start to the end of training.[3]

---

[3]All variants use the same total training budget and the same trajectory buffer; only the checkpoint sampling distribution changes.

*Table 10.* **RQ9: Sampling distribution for TMS** (Qwen-2.5-3B, $T$=10).

| Sampling | MATH ↑ | Cross-task ↓ | ARC-C ↑ | HotpotQA ↑ |
|---|---|---|---|---|
| Early-only | 76.5 | 12.9 | 68.1 | 44.0 |
| Late-only | 80.4 | 9.8 | 70.3 | 45.7 |
| Mid-only | 80.8 | 3.7 | 78.4 | 50.6 |
| Uniform | **81.1** | **2.3** | **79.2** | **51.7** |

For each input $x$, we store one sampled trajectory output $\hat{y}^{(t)}(x)$ per checkpoint $t \in \{1, \ldots, T\}$. During student training, each synthetic target is selected by first sampling a checkpoint index $t \sim p(t)$, then setting $\tilde{y} \leftarrow \hat{y}^{(t)}(x)$ (with the same decoding settings across variants). We compare the following distributions $p(t)$:

- **Early-only:** $p(t) = \frac{1}{\lfloor T/3 \rfloor} \mathbb{1}\{t \leq \lfloor T/3 \rfloor\}$ (uniform over the first $\approx 30\%$ of checkpoints).

- **Mid-only:** uniform over the middle $\approx 30\%$ of checkpoints.

- **Late-only:** $p(t) = \frac{1}{T - \lceil 2T/3 \rceil + 1} \mathbb{1}\{t \geq \lceil 2T/3 \rceil\}$ (uniform over the last $\approx 30\%$ of checkpoints).

- **Uniform:** $p(t) = \frac{1}{T}$ (uniform over all checkpoints).

These choices directly test the "window" hypothesis suggested by mismatch drift: early checkpoints emphasize diversity but are noisier, while late checkpoints emphasize correctness but may reflect over-committed solution modes.

**Windowing matters: early checkpoints are too noisy, late checkpoints can over-specialize.** Table 10 reports results on Qwen-2.5-3B / MATH. Sampling only from early checkpoints substantially degrades target performance, consistent with injecting supervision before the model has learned task format and basic competence. Late-only sampling improves target accuracy but remains much worse than uniform mixing on retention, suggesting that late checkpoints can concentrate around a narrow set of converged trajectories. Mid-only sampling is stronger, but uniform sampling remains the best default because it balances competence and diversity across the trajectory.

> **RQ10 (Effect of $T$ on supervision quality).** *Does increasing the number of checkpoints $T$ improve TMS by capturing more diverse solution modes, or hurt by injecting early noisy supervision, and where is the optimal $T$?*

**Setup (sweeping the number of checkpoints).** We sweep the number of trajectory checkpoints $T$ while holding the *total post-training compute* fixed. Concretely, for each $T \in \{2, 4, 6, 8, 10, 12\}$, we (i) save $T$ equally-spaced checkpoints along the same underlying post-training trajectory (Qwen-2.5-3B on the same target task), (ii) harvest one sampled completion per training example per checkpoint using the same decoding policy, and (iii) train a TMS student under identical optimizer settings and total update steps. Unless otherwise stated, we use **uniform checkpoint sampling** $p(t) = 1/T$ and the same mixture procedure as in Section 5. The reported metrics are Avg Target (mean across the six target tasks) and Cross-task (Avg $\Delta$ on the other target tasks), matching the main-table definitions.

**Moderate $T$ is best: diminishing returns past the mid-range.** Figure 4 summarizes the sweep. Increasing $T$ from small values initially improves Avg Target and reduces cross-task interference, consistent with a richer supervision mixture that (i) better covers alternative valid modes (reducing mode collapse) and (ii) keeps supervision closer to the model's evolving support (reducing mismatch-driven updates). However, beyond a moderate range, gains saturate and variability increases: adding more checkpoints begins to incorporate (a) very early snapshots whose outputs are noisy and format-inconsistent, and (b) very late snapshots that may be over-specialized. Both effects reduce the marginal benefit of additional checkpoints.

In our setting, $T \in [8, 10]$ provides a stable sweet spot for the accuracy–transfer tradeoff. This range captures enough of the trajectory to preserve diversity while avoiding excessive exposure to the lowest-quality early supervision. Table 11 shows that larger $T$ yields diminishing returns while increasing compute and temporary storage. As a practical rule of thumb, we use uniformly spaced checkpoints with $T = 8$–$10$, and exclude or downweight the earliest 10–20% only when early outputs are visibly noisy.

*Table 11.* $T$ **versus cost summary** (Qwen-2.5-3B / MATH).

| $T$ | MATH↑ | ARC-C↑ | FLOPs | Peak storage |
|---|---|---|---|---|
| 4 | 80.2 | 76.9 | 1.6× | 1.20× |
| 8 | 80.9 | 78.8 | 2.0× | 1.36× |
| 10 | 81.1 | 79.2 | 2.2× | 1.45× |
| 16 | **81.2** | **79.3** | 2.8× | 1.72× |

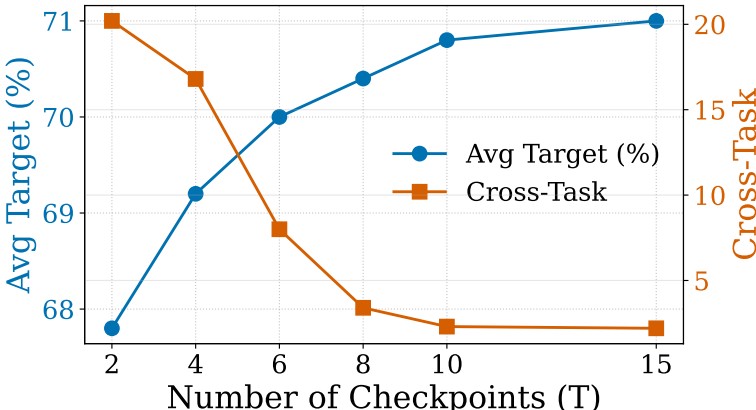

*Figure 4.* **RQ10: Effect of the number of checkpoints** $T$ (Qwen-2.5-3B). We report Avg Target and Cross-task.

### G.3. 70B Scalability Pilot

**TMS trends persist at larger scale.** Table 12 reports a representative LLaMA-3.3-70B-Instruct pilot on reasoning and instruction-following tasks. The same qualitative trend holds: standard SFT improves targets but causes larger retention drops, while TMS remains close to GRPO on retention with similar target performance. The absolute gaps are smaller than in the 1B–8B setting, consistent with larger models being more robust to supervision mismatch.

> **RQ11 (Multimodal feasibility).** *Does trajectory-mixed supervision extend to a multimodal setting, improving retention and reducing drift relative to standard SFT in a single-model pilot?*

> **RQ12 (Agentic feasibility).** *Does TMS extend to a small multi-step tool-use setting, preserving the same KL/PLD–forgetting trends observed in single-turn tasks?*

**Setup.** To test feasibility beyond single-turn text benchmarks, we run two small pilots under the same evaluation lens used throughout the paper: target performance on the pilot task(s) and cross-task interference to the remaining downstream targets. **Agentic pilot:** we fine-tune **Qwen-2.5-7B** on **ToolAlpaca**, a tool-use dataset requiring multi-step tool-augmented responses. **Multimodal pilot:** we fine-tune **Qwen-2.5-Omni-3B** on **TextVQA** and **DocVQA**, which require answering questions about images containing text and scanned documents, respectively. For SFT and TMS, we use the same supervised objective as in Section 5; for GRPO, we use outcome-based rewards when verifiable signals exist and otherwise rely on the benchmark evaluation protocol. Cross-task interference is reported as the average $\Delta$ on the remaining target tasks (defined in Section 7), keeping the measurement consistent with the main results.

**TMS extends to multimodal and tool-use pilots while remaining stable.** Table 13 summarizes the pilot results. Standard SFT substantially improves the agentic and multimodal targets relative to zero-shot but incurs large cross-task interference (29.4), suggesting brittle specialization. In contrast, GRPO achieves minimal interference (1.8), consistent with the general trend that on-policy optimization tends to stay in a low-drift regime. TMS closely tracks this RL-like behavior: it achieves low cross-task interference (2.6) while remaining competitive on both tool-use and multimodal targets.

On the agentic benchmark, TMS nearly matches GRPO (67.9 vs. 68.4 on ToolAlpaca). On multimodal tasks, TMS matches

*Table 12.* **70B scalability pilot** (LLaMA-3.3-70B-Instruct).

| Method | Math500↑ | GSM8K↑ | IFEval↑ | Cross-task↓ | ARC-C↑ | HotpotQA↑ |
|---|---|---|---|---|---|---|
| Base | 91.8 | 94.1 | 81.6 | – | 93.4 | 69.8 |
| Standard SFT | 95.4 | 96.2 | **86.9** | 12.7 | 86.1 | 62.3 |
| REINFORCE | 94.7 | 95.8 | 84.9 | 2.6 | 92.1 | 68.2 |
| GRPO | 95.8 | **96.5** | 85.7 | **1.4** | **93.0** | **69.1** |
| TMS | **95.9** | 96.1 | 86.4 | 2.1 | 92.6 | 68.7 |

*Table 13.* **RQ11–12: Agentic and multimodal pilots** (ToolAlpaca on Qwen-2.5-7B; TextVQA/DocVQA on Qwen-2.5-Omni-3B).

| Method | ToolAlpaca | Multimodal | | Cross-task↓ |
|---|---|---|---|---|
| | | TextVQA | DocVQA | |
| Base | 48.6 | 79.8 | 93.3 | – |
| Standard SFT | 66.2 | 85.1 | 96.2 | 29.4 |
| GRPO | **68.4** | 85.0 | **96.3** | **1.8** |
| TMS (Ours) | 67.9 | **85.2** | 96.2 | 2.6 |

or slightly improves over SFT (TextVQA: 85.2 vs. 85.1) while remaining comparable to GRPO. Taken together, these pilots suggest that trajectory-mixed supervision is not confined to single-turn text tasks; the same "near-policy via trajectory" principle transfers to multi-step tool-use and multimodal post-training, and continues to reduce cross-task interference in the same direction observed in Table 3.

# H. Full Experimental Results

*Table 14.* **Full results across models.** Target-task performance $S_{\text{tgt}}$ (higher is better), cross-task transfer to other target tasks $S_{\text{xfer}}$ (Avg $\Delta$, lower is better), and held-out retention on ARC-C/HotpotQA/SafetyBench. Parentheses denote change vs. the base model; best / second-best are highlighted.

| Method | Target Tasks $S_{\text{tgt}} \uparrow$ | | | | | | Cross-task $S_{\text{xfer}} \downarrow$ | Held-out Retention $\uparrow$ | | |
|---|---|---|---|---|---|---|---|---|---|---|
| | Math500 | MATH | GSM8K | Count. | IFEval | MMLU | Avg $\Delta$ other tgt | ARC-C | Hotpot | SafetyBench |
| **Qwen-2.5-1.5B-Instruct** | | | | | | | | | | |
| Base | 46.6 | 48.1 | 68.9 | 13.2 | 38.3 | 30.8 | – | 66.9 | 43.1 | 35.1 |
| Standard SFT | 58.4 | 62.4 | 77.3 | 25.2 | 54.1 | **39.0** | ↓ 26.2 | 42.3(↓ 24.6) | 19.4(↓ 23.7) | 31.4(↓ 3.7) |
| Self-SFT | 54.9 | 57.1 | 73.1 | 29.4 | 47.3 | 34.7 | ↓ 18.4 | 51.4(↓ 15.5) | 27.5(↓ 15.6) | 34.3(↓ 0.8) |
| Final-SFT | 57.1 | 61.9 | **78.0** | 25.1 | 53.1 | 36.8 | ↓ 24.3 | 48.3(↓ 18.6) | 25.3(↓ 17.8) | 34.0(↓ 1.1) |
| REINFORCE | 56.9 | 60.1 | 75.2 | 26.1 | 49.3 | 34.7 | ↓ 3.4 | 63.8(↓ 3.1) | 40.1(↓ 3.0) | 35.0(↓ 0.1) |
| GRPO | 58.9 | **63.5** | 76.3 | **35.2** | 53.6 | 35.2 | ↓ 1.2 | **66.7**(↓ 0.2) | **42.0**(↓ 1.1) | **36.3**(↑ 1.3) |
| TMS (Ours) | **59.0** | 62.8 | 76.8 | 32.4 | **54.4** | 38.6 | ↓ 2.9 | 65.4(↓ 1.4) | 41.4(↓ 1.7) | 34.8(↓ 0.3) |
| **Qwen-2.5-3B-Instruct** | | | | | | | | | | |
| Base | 60.2 | 62.0 | 85.2 | 29.6 | 60.6 | 43.8 | – | 80.5 | 53.6 | 35.7 |
| Standard SFT | 76.4 | 80.3 | **90.2** | 49.2 | 71.3 | **54.0** | ↓ 39.2 | 42.7(↓ 37.8) | 24.1(↓ 29.5) | 31.1(↓ 4.6) |
| Self-SFT | 74.2 | 77.4 | 87.3 | 48.6 | 67.9 | 47.2 | ↓ 29.3 | 51.8(↓ 28.7) | 33.1(↓ 20.5) | 33.6(↓ 2.1) |
| Final-SFT | 75.8 | 79.6 | 89.5 | 48.1 | 70.2 | 51.4 | ↓ 35.1 | 45.2(↓ 35.3) | 28.4(↓ 25.2) | 33.2(↓ 2.5) |
| REINFORCE | 75.1 | 78.2 | 88.4 | 47.3 | 67.2 | 49.1 | ↓ 4.1 | 78.6(↓ 1.9) | 51.4(↓ 2.2) | **35.5**(↓ 0.2) |
| GRPO | 77.4 | **81.5** | 90.1 | **54.2** | 70.9 | 50.8 | ↓ 1.9 | **80.2**(↓ 0.3) | **53.1**(↓ 0.5) | 35.4(↓ 0.3) |
| TMS (Ours) | **77.8** | 81.1 | 88.9 | 51.3 | **72.0** | 53.6 | ↓ 2.3 | 79.2(↓ 1.3) | 51.7(↓ 1.9) | 35.0(↓ 0.7) |
| **Qwen-2.5-7B-Instruct** | | | | | | | | | | |
| Base | 72.2 | 70.5 | 89.9 | 41.0 | 69.5 | 55.8 | – | 88.8 | 63.9 | 38.3 |
| Standard SFT | 82.6 | 83.4 | **94.8** | 59.4 | **79.7** | **64.2** | ↓ 41.2 | 53.5(↓ 35.3) | 36.2(↓ 27.7) | 34.1(↓ 4.2) |
| Self-SFT | 80.4 | 81.1 | 91.6 | 56.0 | 72.9 | 59.4 | ↓ 33.8 | 68.4(↓ 20.4) | 47.1(↓ 16.8) | 36.9(↓ 1.4) |
| Final-SFT | 81.8 | 82.0 | 92.8 | 58.4 | 75.0 | 63.4 | ↓ 37.0 | 59.8(↓ 29.0) | 43.6(↓ 20.3) | 36.4(↓ 1.9) |
| REINFORCE | 81.1 | 80.4 | 92.1 | 60.1 | 74.1 | 59.0 | ↓ 4.2 | 86.8(↓ 2.0) | 62.1(↓ 1.8) | 38.0(↓ 0.3) |
| GRPO | 83.4 | **85.8** | 93.6 | **69.0** | 77.5 | 60.8 | ↓ 1.6 | **88.4**(↓ 0.4) | **63.4**(↓ 0.5) | **38.6**(↑ 0.3) |
| TMS (Ours) | **83.6** | 83.9 | 93.7 | 62.8 | 78.6 | 62.9 | ↓ 2.8 | 86.9(↓ 1.9) | 62.8(↓ 1.1) | 37.1(↓ 1.2) |
| **LLaMA-3.1-8B-Instruct** | | | | | | | | | | |
| Base | 47.2 | 46.0 | 84.1 | 18.6 | 67.0 | 46.2 | – | 84.1 | 52.6 | 33.8 |
| Standard SFT | 66.7 | 64.6 | **89.6** | 39.2 | 75.2 | **55.1** | ↓ 32.9 | 53.2(↓ 30.9) | 28.2(↓ 24.4) | 30.1(↓ 3.7) |
| Self-SFT | 60.4 | 60.4 | 87.4 | 36.4 | 71.6 | 50.8 | ↓ 24.9 | 64.2(↓ 19.9) | 39.1(↓ 13.5) | 32.6(↓ 1.2) |
| Final-SFT | 64.3 | 63.1 | 88.8 | 37.4 | 73.8 | 53.2 | ↓ 31.4 | 60.8(↓ 23.3) | 36.2(↓ 16.4) | 32.2(↓ 1.6) |
| REINFORCE | 63.9 | 61.8 | 88.1 | 34.8 | 72.2 | 49.3 | ↓ 4.9 | 82.1(↓ 2.0) | 50.6(↓ 2.0) | 33.6(↓ 0.2) |
| GRPO | **70.5** | **65.4** | 89.4 | **41.2** | 75.6 | 50.8 | ↓ 0.9 | **83.8**(↓ 0.3) | **52.1**(↓ 0.5) | **34.1**(↑ 0.3) |
| TMS (Ours) | 66.3 | 64.9 | 89.2 | 40.4 | **76.1** | 54.6 | ↓ 2.3 | 83.1(↓ 1.0) | 51.4(↓ 1.2) | 33.0(↓ 0.8) |
| **LLaMA-3.2-1B-Instruct** | | | | | | | | | | |
| Base | 20.4 | 22.9 | 36.3 | 3.6 | 45.1 | 16.7 | – | 38.9 | 21.9 | 24.8 |
| Standard SFT | 35.2 | 38.8 | 53.1 | 12.8 | 52.8 | **25.2** | ↓ 24.1 | 23.4(↓ 15.5) | 10.8(↓ 11.1) | 21.8(↓ 3.0) |
| Self-SFT | 30.6 | 32.0 | 48.9 | 12.0 | 49.6 | 21.8 | ↓ 17.4 | 29.2(↓ 9.7) | 14.8(↓ 7.1) | 23.6(↓ 1.2) |
| Final-SFT | 33.9 | 37.9 | 51.0 | 12.2 | 51.4 | 23.8 | ↓ 24.6 | 26.8(↓ 12.1) | 13.1(↓ 8.8) | 23.1(↓ 1.7) |
| REINFORCE | 32.4 | 35.6 | 51.4 | 11.9 | 50.1 | 22.1 | ↓ 3.8 | 36.8(↓ 2.1) | 20.1(↓ 1.8) | 24.4(↓ 0.4) |
| GRPO | **38.5** | **39.5** | 54.3 | **15.8** | 42.1 | 24.1 | ↓ 1.6 | **38.2**(↓ 0.7) | 21.2(↓ 0.7) | **25.0**(↑ 0.2) |
| TMS (Ours) | 35.7 | 38.9 | 52.6 | 13.5 | **53.7** | 25.2 | ↓ 3.1 | 37.6(↓ 1.3) | **21.8**(↓ 0.1) | 24.6(↓ 0.2) |

