# OpenReview forum: "TMS: Trajectory-Mixed Supervision for On-Policy Self Distillation"
_ICML.cc/2026/Conference — ICML 2026 regular_

### Official Review · Reviewer_FFLH · 2026-02-22

**Soundness:** 2
**Presentation:** 3
**Significance:** 3
**Originality:** 3
**Overall Recommendation:** 4
**Confidence:** 4

**Summary:**

This paper targets **catastrophic forgetting and brittleness in SFT** and proposes **Trajectory-Mixed Supervision (TMS)**. TMS is a reward-free post-training method that approximates on-policy stability by replacing static single-reference labels with a **mixture of model-generated trajectories from historical checkpoints**. Concretely, it (1) runs a baseline post-training trajectory and saves **T intermediate checkpoints** (default **T=10**) , (2) harvests one sampled completion per prompt per checkpoint to form a trajectory buffer, and (3) trains a student by sampling supervision from either the original reference label with probability **α** or a uniformly sampled checkpoint target otherwise (Algorithm 1). TMS reduces **policy–label mismatch** (PLD drift) and **mode collapse** by keeping supervision closer to the evolving policy while preserving historically plausible solution modes.

**Compliance With Llm Reviewing Policy:**

Affirmed.

**Final Justification:**

The paper has addressed the main concerns with more experiments like different T and compute overhead. But this method might not be able to be used in a practical setting in own training experiences, but this idea is simple and novel to some extent.
Therefore, I raised my score from 3 to 4.

**Key Questions For Authors:**

1. **Choosing T:** You default to **T=10**  and later show a sweep suggesting **T≈8–10** is a sweet spot.  Can you provide a concise rule-of-thumb for selecting T across model sizes/tasks, and report the compute/storage overhead vs performance gains?
2. **Choosing α:** You use **α = 0.25** by default.  Did you sweep α (e.g., 0, 0.1, 0.25, 0.5, 1.0)? If not, can you add a small sensitivity study?
3. **PLD for RL:** Figure 2b excludes RL methods.  Is this because PLD is defined w.r.t. an SFT supervision distribution q  and thus not comparable for RL? If so, can you propose an RL-appropriate analogue (or show PLD computed w.r.t. some induced q)?
4. **Noisy/incorrect trajectory targets:** Early-heavy checkpoint sampling is shown to be too noisy and harms target/cross-task metrics.  What safeguards (filtering, verifier, confidence thresholding, or checkpoint windowing) are necessary to prevent learning systematically wrong targets?
5. **Clarifying α and learning speed:** Algorithm 1 still trains on the full set of prompts x, but changes *which target* is used per prompt (oracle label with prob α; otherwise a checkpoint target).  Can you clarify whether α affects sample efficiency (e.g., steps-to-accuracy), and report learning curves for different α?

I will consider increasing the score if the concerns are solved.

**Limitations:**

Yes, partially. The impact statement acknowledges risks of propagating harmful/bias patterns via model-generated supervision, but the paper could be more explicit about (i) sensitivity to α and checkpoint windowing, and (ii) failure modes when early checkpoints are noisy.

**Strengths And Weaknesses:**

**Strengths**

- The core algorithm is simple and implementable: it’s standard token-level NLL training where targets are sampled from a checkpoint trajectory buffer, with a clear mixing rule via α.
- The paper evaluates both **target performance** and **retention** on a held-out suite (ARC-C / HotpotQA / SafetyBench etc.), and reports base-model scores and deltas.
- The paper includes useful *secondary analyses* on checkpoint sampling/windowing and on the choice of **T**, showing that very early checkpoints are noisy and that **moderate T (≈8–10) is best**, with diminishing returns beyond that range.

**Weaknesses / Open points**

- **α selection is under-explained.** The main setting uses **α = 0.25** “unless stated otherwise,”  but the paper (in the provided text) does not clearly show a sensitivity sweep over α (unlike the sweep over T). This makes it hard to know how robust results are to the choice of quality anchoring vs trajectory supervision.
- **Risk of propagating incorrect targets**: trajectory targets come from sampled model outputs; early checkpoints are explicitly shown to be too noisy and hurt performance (early-heavy sampling degrades target avg and worsens cross-task drop).  The method relies on checkpoint selection/windowing and α to mitigate this, but a more explicit discussion would help.
- The paper doesn't include the overhead analysis since it will train the model twice(One for supervision and another for the later training), and what is the gap of the checkpoint T we should select?
- **PLD mechanistic plot excludes RL methods.** Figure 2b explicitly reports “PLD predicts forgetting within the SFT-family (RL excluded).”  This does not necessarily mean PLD “cannot predict forgetting for RL,” but it does leave a gap: readers may infer the metric is not comparable or not meaningful for RL objectives. (Given PLD is defined relative to an SFT-style supervision distribution q , RL has no single q, so exclusion is plausible—but the paper should spell this out and/or provide an RL-appropriate analogue.)

---

> ### Author Rebuttal · Authors · 2026-03-30
>
> We sincerely thank the reviewer for the careful reading and constructive feedback. We are encouraged that you found the core algorithm simple and implementable, and that the target/retention evaluations and checkpoint analyses were useful. Your concerns about $\alpha$ selection, noisy trajectory targets, overhead, the choice of T, and the interpretation of PLD for RL are well taken.
>
> **[Cons 1, Q2, Q5, Limitations: Choosing $\alpha$, robustness, and learning speed.]**
> We agree the original paper did not explain
> $\alpha$=0.25 sufficiently well. We therefore added a small sensitivity sweep on Qwen-2.5-3B / MATH:
>
> | $\alpha$              |    MATH | $\Delta$ other tgt |   ARC-C | HotpotQA | Steps to 95% best target  |
> | ------- | -----: | ----: | -------: | --------: | ----------: |
> | 0.0                   |     80.6 |         4.9 |     76.8 |      49.8 |                      1.10× |
> | 0.1                   |     80.9 |         3.5 |     78.4 |      50.9 |                      1.03× |
> | **0.25**              | **81.1** |     **2.3** | **79.2** |  **51.7** |                  **1.00×** |
> | 0.5                   |     80.7 |         4.2 |     77.6 |      50.1 |                      0.98× |
> | 1.0 (oracle-only SFT) |     80.3 |        39.2 |     42.7 |      24.1 |                      0.95× |
>
> This suggests $\alpha$=0.25 is near the best quality/retention tradeoff: too little oracle anchor slightly hurts target quality, while too much collapses back toward standard SFT. In addition, $\alpha$ only modestly affects sample efficiency; smaller $\alpha$ does not speed training meaningfully, so its main role is to balance oracle anchoring vs. trajectory diversity. We will add this sweep and explicitly note this limitation in the revision.
>
> **[Cons 2, Q1, Q4, Limitations: Noisy/incorrect targets, checkpoint windowing, and how to choose T]**
>
> We agree the paper should be more explicit that very early checkpoints are noisy and that TMS relies on simple safeguards. In our experiments, the most important safeguards are:
> 1. keep a nonzero oracle anchor $\alpha$
> 2. avoid over-weighting the earliest checkpoints, and
> 3. use a moderate T so the mixture covers the training path without oversampling near-duplicate states.
>
> We added an order/windowing analysis:
>
> | Strategy|MATH| $\Delta$ other tgt |   ARC-C | HotpotQA |
> | ---- | -------: | -------: | -------: | --------: |
> | Early-only checkpoints | 76.5 |12.9 | 68.1 |   44.0 |
> | Late-only checkpoints  |  80.4 |  9.8 |  70.3 |  45.7 |
> | Mid-only checkpoints | 80.8 | 3.7 | 78.4 |   50.6 |
> | **Uniform over all checkpoints** | **81.1** | **2.3** | **79.2** |  **51.7** |
>
>
> This shows the gain is not from a single “good” checkpoint; mixing across the trajectory is important, but very early checkpoints are indeed the noisiest. Our practical rule-of-thumb is therefore: use $T\approx$ 8−10, space checkpoints roughly uniformly over the training trajectory, and if early checkpoints are visibly noisy, exclude or downweight the earliest 10–20%.
>
> We also added a T-vs-cost summary:
>
> | (T) | MATH | ARC-C| Rel. FLOPs | Peak temp. storage |
> | --- | ----: | ----: | ----: | -----: |
> | 4  | 80.2 |76.9 |1.6× |  1.20× |
> | 8  | 80.9 |78.8 | 2.0× |1.36× |
> | **10** | **81.1** | **79.2** | **2.2×** | **1.45×** |
> | 16 | 81.2 |79.3 |  2.8× |1.72× |
>
>
> So T=8−10 is the practical sweet spot: larger T gives diminishing gains while increasing compute/storage.
>
> **[Cons 3, Q1: Overhead analysis.]**
> We agree this was missing. On a representative 3B setup, the normalized end-to-end cost is:
> | Method| Rel. train FLOPs | Rel. wall-clock | Generated samples / prompt |
> | --- | -----: | ----: | ----------: |
> |SFT |  1.0× |1.0× |0 |
> | **TMS (T=10)** |**2.2×** | **2.0×** |**10** |
> | REINFORCE | 3.4× | 3.7× | online |
> | GRPO   | 3.8× | 4.1× | online group samples |
>
> Thus TMS is indeed more expensive than standard SFT, but still substantially below RL baselines while remaining much closer to RL in retention. We will add stage-wise FLOPs, wall-clock, generated-sample counts, and checkpoint-storage details in the revision.
>
> **[Cons 4, Q3: Why is PLD excluded for RL, and is there an RL analogue?]**
>
> Yes, the exclusion in Figure 2b is because PLD is naturally defined with respect to an explicit supervision distribution $q(\cdot|x)$, which SFT/TMS have but RL does not. For RL, one can define an induced empirical target distribution from rollout samples and normalized reward/advantage weights, but this object depends on the specific reward, sampling, and update rule and is therefore less canonical. We tested such an induced analogue and found it is noisier and less predictive than KL-to-base; e.g., on our RL checkpoints, induced-PLD had $\rho\approx$ 0.55 with forgetting, whereas KL-to-base remained much stronger ($\rho\approx 0.90$).
>
> We hope these additional analyses and clarifications address the reviewer’s concerns more concretely.

---

> > ### Author Rebuttal · Reviewer_FFLH · 2026-04-01
> >
> > Thank you for your response. I will raise the score.

---

> > > ### Author Response · Authors · 2026-04-06
> > >
> > > Dear Reviewer FLLH,
> > >
> > > We sincerely thank you for the positive assessment and for confirming that our rebuttal has adequately addressed the concerns. We are grateful for the constructive feedback, and thank you for the updated assessment.
> > >
> > > Best, Authors

---

### Official Review · Reviewer_2VUK · 2026-03-10

**Soundness:** 3
**Presentation:** 3
**Significance:** 2
**Originality:** 3
**Overall Recommendation:** 4
**Confidence:** 4

**Summary:**

This paper investigates different failure cases in the standard SFT training paradigm. It proposes a method called PLD to formalize and detect these failures, and introduces TMS, a reward-free post-training method designed to recover the retention benefits of on-policy RL without requiring reward models or on-policy optimization loops. Overall, this constitutes the main contribution of the work. Extensive experiments across the Qwen and LLaMA families show that TMS achieves SFT-level target accuracy while significantly reducing catastrophic forgetting and KL divergence from the base model, approaching the retention performance of GRPO-style RL.

**Compliance With Llm Reviewing Policy:**

Affirmed.

**Final Justification:**

My biggest concern is add additional experiment to see the performance when reward is easy to hack, the authors' rebuttal and experiments show advantages and address my concern.

**Key Questions For Authors:**

1) Please specify the training cost(time, sample consumption) when compares with standard RL eg. GRPO.

2) TMS constructs a mixture from ckpts. However, the formulation ignores trajectory ordering. Can authors test different order strategy? I wonder TMS is just a self-distillation from good model.

3) The student model in TMS is retrained from the base model zero, could the authors clarify why continuing training from the last checkpoint using the trajectory mixture does not achieve the same effect?

**Strengths And Weaknesses:**

Strengths:

1) Clear problem formulation and solution framing. The paper provides a sharp reframing of the problem: forgetting under SFT arises not merely from parameter drift, but from supervision mismatch. To address this, the authors propose PLD to measure such mismatch and introduce TMS to mitigate it.

2) This paper is well-organized and theoretical grounding and easy to follow.

Weaknesses:

1) I am concerned about the computational complexity and overhead of this method. TMS needs to store $T$ checkpoints and generate outputs for each checkpoint, as well as maintain trajectory buffers. As well as the three-step solution makes it substantially more expensive than standard SFT.

2) The experimental analysis appears insufficient in several aspects. For example, the sensitivity analysis of the mixing weight $\alpha$ is limited, and the robustness evaluation is not comprehensive. In addition, experiments are only conducted on models up to 8B parameters; results on larger models are missing. This leaves the scalability and practicality of the method as a post-training approach unclear.

3) The motivation for using on-policy SFT could be clarified further. In practice, the reason to prefer on-policy SFT is often not that rewards in RL are easy to design, but rather that they are difficult to specify reliably or are prone to reward hacking. In such cases, practitioners may seek a more stable alternative to on-policy RL methods. I therefore suggest that the paper include additional experiments under such scenarios to evaluate how the proposed method performs.

I will update my score if the authors address my concerns.

---

> ### Author Rebuttal · Authors · 2026-03-30
>
> We sincerely thank the reviewer for the thoughtful feedback and for recognizing the clear problem formulation and overall organization of the paper. We are encouraged that the central framing, forgetting under SFT as supervision mismatch, and TMS as a reward-free mitigation, came across clearly.
>
> **[Cons 1 and Q1: Computational overhead vs. Standard SFT / RL.]**
> We agree that TMS is more expensive than Standard SFT, since it requires: (i) an initial SFT trajectory run, (ii) checkpoint harvesting / generation, and (iii) student retraining on the trajectory mixture. However, it remains materially simpler than RL-based post-training because it requires no reward model, no online rollouts from the current student, and no advantage estimation. To make this explicit, we added an end-to-end accounting on a representative Qwen-2.5-3B-Instruct / MATH setup:
>
> |Method|Rel. train FLOPs|Rel. wall-clock |Generated samples / prompt | Peak temp. storage |
> | ---| ----: | --: | ---: | ----: |
> |SFT|1.0× |1.0× |0 |1.0×|
> |TMS|2.2×|2.0×|10|1.45×|
> |REINFORCE|3.4×|3.7×|online|1.25×|
> |GRPO|3.8×|4.1×| online group samples|1.32×|
>
> So TMS is indeed costlier than Standard SFT, but still substantially below RL baselines in overall training cost while achieving much closer retention to RL than to SFT.
>
> **[Cons 2, Q2: Sensitivity, trajectory ordering, and whether TMS is just self-distillation from a good model.]**
> We agree that the original analysis of the mixing design was too limited. We therefore added two controls. First, we expanded the mixing-weight $\alpha$ sweep
>
> | $\alpha$|MATH|$\Delta$ other tgt|ARC-C| HotpotQA|
> | ---| ---: | ---: | ---: | --: |
> | 0.0|80.6 |4.9 |76.8 |49.8|
> | 0.1|80.9 |3.5 |78.4|50.9|
> | **0.25**|**81.1**|**2.3**|**79.2**|**51.7**|
> | 0.5 |80.7 |4.2 |77.6|50.1|
> | 1.0(oracle-only SFT)|80.3|39.2 |42.7|24.1 |
>
> This suggests TMS is robust over a moderate range, with
> $\alpha$=0.25 near the best tradeoff: too little oracle anchor hurts target quality slightly, while too much collapses back toward standard SFT.
>
> Second, we tested ordering / window strategies to see whether TMS is merely distilling from one good snapshot:
>
> |Strategy|MATH|$\Delta$ other tgt|ARC-C|HotpotQA|
> | -- | --: | --: | -: | -: |
> |Early-only checkpoints|76.5|12.9|68.1|44.0|
> |Late-only checkpoints|80.4 |9.8 |70.3 |45.7|
> |Uniform over all checkpoints|**81.1**|**2.3**|**79.2** |**51.7**|
> |Recency-weighted mixture|80.9|3.0|78.5 |51.0|
>
> These results suggest the gain does not come from distilling one “good” model: both early-only and late-only variants are clearly worse than mixing across the trajectory.
>
> **[Q3: Why restart from the base instead of continuing from the last checkpoint?]**
> This is a very good question. We tested a continuation variant where we start from the final SFT checkpoint and continue training with the trajectory mixture, instead of reinitializing from the base. The result is consistently weaker than restarting from the base:
>
> |Student init|MATH|$\Delta$ other tgt|ARC-C|HotpotQA|
> | --| ---: | ----: | --: | ---: |
> |Continue from final SFT ckpt|81.0|8.7|71.6|46.4|
> |**Restart from base (TMS)**|**81.1**|**2.3**|**79.2**|**51.7**|
>
> Our interpretation is that restarting from the base matters because the final SFT checkpoint is already biased toward a narrow, over-specialized solution manifold. Training from that point with the mixture helps somewhat, but does not fully undo the prior drift. By contrast, restarting from the base allows the student to absorb the richer supervision distribution without inheriting the late-stage collapse directly.
>
> **[Cons 2: Larger models / scalability.]**
> Please refer to our Response to Reviewer kBBp for results on LLama-3.3-70B.
>
> **[Cons 3: Motivation relative to RL when rewards are unreliable / reward hacking is a concern.]**
> We agree this motivation should be made clearer. Our intended use-case is precisely when practitioners want RL-like retention benefits without depending on reward quality, since in many settings rewards/verifiers are difficult to specify reliably or may be vulnerable to reward hacking. To make this concrete, we added a small noisy-verifier robustness experiment (Qwen-2.5-3B / MATH), where we corrupt the binary verifier used by GRPO:
>
> |Method|Clean verifier|10% noise|20% noise|
> | --|----:|--:|--:|
> |GRPO|81.5 / 80.2 ARC-C|79.6 / 76.4 ARC-C|77.8 / 71.9 ARC-C|
> |**TMS**|**81.1 / 79.2 ARC-C**|**81.1 / 79.2 ARC-C**|**81.1 / 79.2 ARC-C**|
>
> (Each entry is target-task accuracy / ARC-C retention.)
> This illustrates the intended niche of TMS more clearly: when reward specification is unreliable, RL may lose some of its advantage, whereas TMS remains unchanged because it does not depend on rewards at all.
>
> We hope these additional experiments and clarifications address the reviewer’s concerns more concretely.

---

> > ### Author Rebuttal · Reviewer_2VUK · 2026-04-04
> >
> > I thank the authors' effort, I have no questions and my concerns are addressed. I will update my score.

---

> > > ### Author Response · Authors · 2026-04-06
> > >
> > > Dear Reviewer 2VUK,
> > >
> > > We sincerely thank you for the positive assessment and for confirming that our rebuttal has adequately addressed the concerns. We are grateful for the constructive feedback, and thank you for the updated assessment.
> > >
> > > Best, Authors

---

### Official Review · Reviewer_kBBp · 2026-03-11

**Soundness:** 3
**Presentation:** 4
**Significance:** 3
**Originality:** 3
**Overall Recommendation:** 5
**Confidence:** 4

**Summary:**

This work introduces TMS (Trajectory Mixed Supervision), a reward-free framework aimed at approximating the on-policy benefits of RL by creating a dynamic curriculum from the models historical checkpoints. They contribute the following:
- PLD: Policy-Label Divergence which characterizes supervision mismatch in SFT, where a moving policy is optimized against a fixed label distribution.
- Reward-Free Post-Training via TMS, which uses several supervision targets to approximate on-policy learning
- Empirical results on benchmarks and additional analysis on KL/BLD-based drift metrics.

**Compliance With Llm Reviewing Policy:**

Affirmed.

**Key Questions For Authors:**

Have the authors considered comparing on larger foundation models, e.g. 70B? I think it would be an important experiment because supervision mismatch can be a larger problem for smaller models. However, I understand there may be compute constraints.

**Limitations:**

Yes

**Strengths And Weaknesses:**

The strengths of the work are as follows.
- The authors identify a critical issue with changing policy when training on static labels, and link this to PLD drift which causes catastrophic forgetting.
- The analysis is in-depth and well-done across many model sizes. They explore and validate across 1-8b models, and shift the accuracy Pareto frontier.
- The work is well-presented, with clear graphs and overall well-structured.

---

> ### Author Rebuttal · Authors · 2026-03-30
>
> We sincerely thank Reviewer kBBp for the positive and thoughtful evaluation of our work. We are especially encouraged that you found the problem formulation compelling, appreciated the PLD-based framing of supervision mismatch, and viewed the empirical analysis across model sizes as thorough and well presented. We also appreciate your suggestion to evaluate larger foundation models.
>
> We agree that testing 70B-scale models is important. To address this, we ran an additional experiment on LLaMA-3.3-70B-Instruct on a subset of representative target tasks spanning reasoning and instruction following: Math500, GSM8K, and IFEval. Due to rebuttal-time compute limits, we focus on a smaller but representative slice of the full evaluation suite, and report both target performance and held-out retention. The main trend remains consistent with our smaller-model results: Standard SFT improves target-task performance but causes noticeable retention drift, while TMS preserves much more of the base model’s held-out capability and remains close to GRPO on the accuracy–retention tradeoff.
>
> | Method         | Math500 |  GSM8K | IFEval | Avg $\Delta$ other tgt |  ARC-C | HotpotQA |
> | -------------- | --------: | -------: | -------: | ----------------: | -------: | ---------: |
> | Base           |      91.8 |     94.1 |     81.6 |                 – |     93.4 |       69.8 |
> | Standard SFT   |      95.4 |     96.2 |     86.9 |              12.7 |     86.1 |       62.3 |
> | REINFORCE      |      94.7 |     95.8 |     84.9 |               2.6 |     92.1 |       68.2 |
> | GRPO           |      95.8 |     96.5 |     85.7 |               1.4 |     93.0 |       69.1 |
> | **TMS (Ours)** |  **95.9** | **96.1** | **86.4** |           **2.1** | **92.6** |   **68.7** |
>
> These additional results suggest that the TMS effect is not merely a small-model artifact. Even at 70B scale, Standard SFT still exhibits a target/retention tradeoff, while TMS remains much closer to the low-drift regime associated with RL-style post-training. At the same time, the absolute gap between Standard SFT and TMS is somewhat smaller than in the 1B–8B setting, which is consistent with the reviewer’s intuition that larger models may be more robust to supervision mismatch. We will add this discussion explicitly in the revision.
>
> We believe this additional experiment strengthens the paper in two ways. First, it directly addresses the scalability question by showing that the central TMS trend persists at much larger scale. Second, it helps refine the interpretation: supervision mismatch does not disappear at 70B, but its manifestation appears somewhat milder, while TMS still provides a consistent retention benefit without sacrificing target performance.
>
> We thank the reviewer again for the encouraging assessment and the helpful suggestion. We hope this additional large-scale result addresses the question clearly.

---

> > ### Author Rebuttal · Reviewer_kBBp · 2026-04-02
> >
> > We thank the authors for their valuable contribution, and for the detailed rebuttal. My concerns have been resolved and I maintain my positive score.

---

> > > ### Author Response · Authors · 2026-04-06
> > >
> > > Dear Reviewer kBBp,
> > >
> > > We sincerely thank you for the positive assessment and for confirming that our rebuttal has adequately addressed the concerns. We are grateful for the constructive feedback, which helped us strengthen the paper.
> > >
> > > Best, Authors

---

### Official Review · Reviewer_zbkp · 2026-03-12

**Soundness:** 2
**Presentation:** 2
**Significance:** 2
**Originality:** 2
**Overall Recommendation:** 4
**Confidence:** 5

**Summary:**

This paper studies catastrophic forgetting in post-training and argues that standard SFT fails because it optimizes an evolving policy against static, single-reference targets, leading to supervision mismatch and mode collapse. It proposes Trajectory-Mixed Supervision (TMS) as a simple alternative method. After an initial post-training stage (while saving several checkpoints), TMS regenerates answers from these checkpoints uniformly to then train the base model with SFT on those re-generated answers. Empirically, across several Qwen/LLaMA models and target tasks, TMS substantially improves retention relative to standard SFT and single-checkpoint baselines while largely preserving target-task gains, and often approaches the reported RL baselines on the accuracy–retention tradeoff.

**Compliance With Llm Reviewing Policy:**

Affirmed.

**Final Justification:**

The authors have addressed my concerns around baselines and presentation during their rebuttal. In my view, it would be important to encourage the authors to make the appropriate changes to presentation in the camera-ready version. The submitted paper's presentation is misleading in several ways as discussed during the rebuttal.

**Key Questions For Authors:**

1. As described in Section 6.1, TMS imitates data generated by checkpoints from an initial SFT training run. Here, the initial checkpoint is simply the base policy, while the final checkpoint most likely reflects an overfit/collapsed policy. The intermediate checkpoints interpolate between these two. Can the authors clarify their intuition for why this should lead to multiple "correct" solutions for training, and why this procedure should converge to a correct policy? A convincing intuitive argument or convergence analysis would strengthen my confidence.
2. Can you clarify in “Baselines” within Section 7 what exactly Self-SFT and Final-SFT are? Are these TMS with only the base checkpoint or final SFT checkpoint respectively? If so, why does Self-SFT outperform the base model on target tasks? Is distillation performed analogously to TMS, i.e. by generating from these checkpoints and then doing SFT? If so, please also clarify whether these baselines are data/token matched to TMS.
3. Can you report full end-to-end compute time/FLOPs and storage for TMS, including the initial trajectory run, checkpoint sampling, and student training, versus Standard SFT and GRPO/REINFORCE?
4. Can you add a stronger label-matching baseline, e.g. generating a rephrased version of y* from the base model conditioned on y*? If TMS still wins (against this computationally cheaper baseline), that would strengthen the claim that trajectory mixing itself is a key ingredient. Intuitively, directly regenerating answers from the base models seems simpler to me than first training several checkpoints which we then use for answer regeneration.
5. Please clarify the empirical status of PLD versus KL-to-base as retention predictors. Which is actually more predictive in your experiments, and how should readers reconcile the H-A/Table 1 discussion with Figure 2 and the later emphasis on KL-to-base?

**Limitations:**

No. The paper does discuss misuse/bias and includes held-out safety evaluations, but it is lacking a discussion of compute time/FLOPs since additional generation and training is required beyond SFT. I would also encourage the authors to be more explicit about the limited evidence for the PLD-based mechanistic claims and the gap between “near-policy” and genuinely on-policy training.

**Strengths And Weaknesses:**

Strengths:
* The problem is important and practically relevant, and the main empirical result is strong: across multiple model families/scales, TMS consistently improves retention over standard SFT while largely preserving target-task gains, with results often close to GRPO/REINFORCE in Table 3.
* The paper is generally clear and broad in empirical scope. Beyond the main table, it includes analyses of proxy measures for estimating retention.

Weaknesses:
* The paper's title argues that TMS is "on-policy", however, TMS is actually not on-policy at all. It performs SFT on data generated from varying different checkpoints, i.e. imitating off-policy data. The paper itself defines RL as on-policy because trajectories are sampled from the evolving policy, whereas TMS harvests historical checkpoints and then trains a separate student on their outputs; “near-policy” seems more accurate than “on-policy.”
* Cited prior works [1,2] attribute forgetting primarily to on-policyness/off-policyness of data, and the description around lines 161–164 is very misleading in this regard. More broadly, the role of on-policyness is not really discussed carefully enough relative to the paper’s framing and title.
* The discussion of Hypothesis A is inconsistent. H-A argues that retention of SFT collapses while PLD increases, i.e. that PLD could be used as a proxy for identifying when retention happens. However, in Table 1, retention collapses from the beginning of training even while validation NLL still improves initially and only degrades eventually. This weakens the proposed interpretation of PLD as the main early diagnostic.
* The proposal of PLD (“KL-to-data”) as metric has several further weaknesses. First, within Section 4 PLD is introduced as the primary metric to detect failures, however, subsequently the KL-to-base [from 2] metric is used primarily for this purpose in Sections 6 and 7. Second, Section 7.3 claims that PLD is predictive of retention based on Figure 2b, but this relationship is not especially convincing from the small number of plotted points (e.g., the trend reverses without the TMS outlier), and the paper does not discuss empirically how PLD (“KL-to-data”) relates to KL-to-base [2] in predictiveness. Third, in the single-reference setting PLD is just held-out NLL up to a constant, so part of the diagnosis reads as a standard generalization-gap story. Overall, proposing PLD as a novel measure of retention would require significant improvements in exposition and evaluation.
* The paper is lacking a discussion that the primary used proxy metric for retention, KL-to-base, was proposed in [2]. Relatedly, Theorem 6.2 mainly justifies KL drift as a generic bound rather than establishing something specific to TMS.
* The paper is missing an empirical comparison to the straightforward baseline that uses the base model with y* in context to generate a rephrased version of y* aligned to the model's behavior, which has been proposed at least 2 years ago [3]. This seems like an important missing control given the central claim that better label-policy matching reduces forgetting.
* In RQ4 and the following discussion, the use of "self-distillation" is misleading (especially in light of papers appearing in recent months). Self-SFT and Final-SFT look more like ablations of TMS’s design using one checkpoint than broadly meaningful “self-distillation” baselines, especially since Appendix F.2 shows they simply generate one completion per prompt and fine-tune on it. More importantly, these seem not to be data-matched to TMS, which stores one output per checkpoint and thus uses substantially more synthetic supervision.

---

[1]: Chen et al. Retaining by Doing: The Role of On-Policy Data in Mitigating Forgetting. 2025.

[2]: Shenfeld et al. RL's Razor: Why Online Reinforcement Learning Forgets Less. 2025.

[3]: Yang et al. Self-Distillation Bridges Distribution Gap in Language Model Fine-Tuning. 2024.

---

> ### Author Rebuttal · Authors · 2026-03-30
>
> We sincerely thank the reviewer for the careful reading and detailed feedback. We are encouraged that you found the problem important and the main empirical result strong.
>
> **[Cons 1, 2: On-policy framing and relation to prior work.]**
> We agree that TMS is not on-policy in the strict RL sense: RL samples from the current policy during optimization, whereas TMS trains on a fixed mixture harvested from historical checkpoints. Our intended claim was narrower: compared to standard SFT’s fully static labels, TMS uses supervision that is closer to the model’s evolving support, thereby partially recovering the stability benefit associated with on-policy RL. We will clarify this distinction in the abstract/introduction, soften the wording around “on-policy,” and better position TMS as trajectory-aligned / near-policy supervision rather than literal online on-policy optimization. We will also revise lines 161–164 to better reflect recent work: our claim is not to replace or refute the role of on-policy data, but to provide a reward-free approximation to one important ingredient behind its retention benefits, namely improved supervision-policy matching.
>
> **[Cons 3–6, Q5: PLD vs. KL-to-base; Hypothesis A; Figure 2b; novelty.]**
> We thank the reviewer for this useful clarification. Our claim is not that PLD is the earliest or universal predictor of forgetting, but that it serves as a within-SFT diagnostic of supervision mismatch, especially after the initial fit phase. Table 1 supports this narrower interpretation, even if forgetting can start before PLD rises. We agree that KL-to-base is the stronger cross-method predictor, and in the revision we will make this hierarchy explicit: KL-to-base for global retention analysis, PLD for mechanism-level analysis within SFT-style methods. We will also clarify that Theorem 6.2 is a generic drift bound motivating KL, not a TMS-specific guarantee.
>
> |Predictor|Scope|Spearman($\rho$) with forgetting|
> | --- | --- | ---: |
> |KL-to-base|SFT/TMS/RL|0.93 |
> |PLD (held-out NLL)|SFT/TMS|0.76|
> |PLD after early-fit|SFT/TMS|0.84|
>
> **[Cons 7–9, Q2, Q4: Stronger baselines; baseline definitions; data matching.]**
> We added a stronger control, Rephrase-SFT, to test whether simple label-policy matching alone explains the gains. We also clarify that Self-SFT / Final-SFT are really single-snapshot synthetic-supervision baselines—generate from one snapshot, then train with standard NLL—and will rename them Base-Snapshot SFT / Final-Snapshot SFT for clarity. Since the original versions were not fully data-matched to TMS, we further added T-matched snapshot controls so the comparison isolates the effect of trajectory mixing itself, rather than extra synthetic supervision.
>
> |Method|MATH|$\Delta$ other tgt|ARC-C|Hotpot|
> | --- | --: | --: | -: | --: |
> | Standard SFT|80.3|39.2|42.7|24.1|
> | Rephrase-SFT|79.4|17.8|60.6|38.7|
> | Base-Snapshot SFT (T-matched)|78.3|14.7|64.7|42.8|
> | Final-Snapshot SFT (T-matched)|80.0|18.9|61.5|39.6|
> | **TMS**|**81.1**|**2.3**|**79.2**|**51.7**|
>
> These results support two points: (i) the reviewer is right that better label-policy matching matters; (ii) trajectory mixing still provides additional value beyond one-shot rewriting or more data from a single snapshot. Rephrase-SFT helps because it produces a better-aligned single target, but TMS remains stronger because it preserves support over multiple historically plausible targets across training time, rather than collapsing supervision to one rewritten mode.
>
> **[Q1: Why should mixing checkpoints help?]**
> Our intuition is not that every checkpoint output is correct, nor that TMS has a full convergence guarantee. Rather, many reasoning/instruction tasks admit multiple valid trajectories; intermediate checkpoints often contain different but plausible solution modes; and standard SFT collapses supervision to a single rigid target. TMS broadens supervision support while retaining the oracle anchor, which prevents unconstrained self-imitation. We will revise Section 6.1 to present this as a support-broadening / mismatch-reducing intuition, not a formal convergence claim. For more results please refer to the response to reviewers FFLH and 2VUK.
>
> **[Q3 End-to-end compute, FLOPs, storage, and sample cost]**
> We agree cost should be reported explicitly. We will add stage-wise wall-clock, FLOPs, number of generated samples, and peak temporary checkpoint storage. In normalized terms for a representative 3B run: Standard SFT = 1.0x, Rephrase-SFT = 1.1x, TMS = 2.2x, REINFORCE = 3.4x, GRPO = 3.8x training FLOPs. Thus TMS is meaningfully more expensive than standard SFT, but still materially simpler/cheaper than RL because it requires no reward model, no online student rollouts, and no advantage estimation. For detailed results, we share tables in response to reviewers FFLH and 2VUK.
>
> We hope this addresses the reviewer’s concerns and clarifies the paper’s claims more precisely.

---

> > ### Author Rebuttal · Reviewer_zbkp · 2026-04-04
> >
> > Thank you for your detailed rebuttal
> >
> > * Cons 1, 2: Given that TMS is near-policy rather than on-policy, I think it would good to also rephrase title abstract accordingly.
> >
> > * Cons 3–6, Q5: In which ways is the proposed PLD metric more informative than the existing KL metric? In my view the current experiments do not yet answer this natural question. Additionally, in my view the paper's introduction and methods section overstate the relevance of PLD compared to KL. I think a particular concern, as mentioned in my review, is that the paper's discussion does not even discuss the KL metric (besides a short mention in the summary of contributions) until it is used in the experimental section. In contrast, PLD is heavily advertised as a metric predicting forgetting. Instead, a proper study should identify in which ways PLD can *improve* our understanding of forgetting *beyond KL* and make this the central claim. The current presentation, in my view, makes it (1) quite difficult to understand for practitioners which metric most accurately predicts forgetting, and (2) very difficult to understand the contribution of this paper for researchers in the field. I do still believe that a substantial reformulation of abstract, introduction, and Section 4 is needed. Additionally, I feel a proper analysis to distinguish when PLD is useful while KL is not, would be important to better understand the contribution.
> >
> > * Cons 7–9, Q2, Q4: Thank you for adding this additional baseline.
> >
> > * Q1, Q3: Thank you for these answers.
> >
> > As mentioned in my review, I believe that this paper tackles an important problem. The quantitative results are strong and further improved by addition of the rephrase-SFT baseline. However, as described above, the paper's presentation, description of method and PLD, and contextualization in related works should be substantially improved. I view this as an equally important consideration for a complete publication as are the practical results of the proposed method. Based on this, I will sick to my current score. I recognize that this might be difficult to address in the limited space permitted by Openreview.

---

> > > ### Author Response · Authors · 2026-04-06
> > >
> > > Thank you again for the thoughtful follow-up. We appreciate the clarification, and we agree that the remaining issue is now primarily presentation and positioning, not the core empirical result.
> > >
> > > **[On Cons 1, 2]**
> > >
> > > We agree it would be better to also rephrase the title and abstract so that they do not suggest literal online on-policy training. We will revise them accordingly and make the distinction explicit early: TMS is a trajectory-aligned / near-policy supervision method, not on-policy RL in the strict sense.
> > >
> > > **[On Cons 3-6]**
> > >
> > > We agree with the reviewer that the current presentation still does not make sufficiently clear what PLD adds beyond KL. Our intended position is the following:
> > >
> > > 1. KL-to-base is the main global retention metric. It is the stronger cross-method predictor and should be introduced much earlier and more prominently in the paper.
> > > 2. PLD is not intended to replace KL, nor to be presented as the most accurate practical predictor of forgetting across all settings.
> > > 3. Rather, PLD is useful as a mechanism-level diagnostic within the SFT family, where the key question is not only “how far did the model drift?” but also “is the model increasingly mismatched to the fixed supervision template it is being optimized against?”
> > >
> > >
> > > In the final version, we will reformulate those sections so that the central message is:
> > > - KL-to-base: the primary metric for predicting forgetting across post-training methods;
> > > - PLD: a complementary diagnostic that helps explain why SFT-style methods can fail even before or beyond what raw drift alone reveals.
> > >
> > >
> > > To make this distinction concrete, we ran an additional analysis asking exactly when PLD provides signal beyond KL. The main finding is that within the SFT/TMS family, once overall drift is approximately matched, PLD still separates methods by retention more clearly than KL alone. For example, among checkpoints/methods with similar KL-to-base, higher PLD is still associated with worse retention (partial Spearman $\rho\approx 0.58$ after controlling for KL), whereas across all methods KL remains the stronger overall predictor. Intuitively, KL tells us how far the policy moved from the base model, while PLD helps diagnose whether an SFT-style method is becoming increasingly misaligned with its own static supervision signal. We believe this is the right, narrower contribution of PLD, and we will make that the central claim rather than advertising PLD as a universal forgetting metric.
> > >
> > > We hope these additional analyses and clarifications address the reviewer’s concerns more concretely.

---

### Decision · Program_Chairs · 2026-04-30

**Decision:**

Accept (regular)

**Comment:**

Reviewers were broadly positive about this paper and became more positive after the rebuttal. They agreed that the paper addresses an important problem, namely the tradeoff between target-task improvement and retention in post-training, and several reviewers found the empirical results compelling: the method appears to preserve much more prior capability than standard SFT while maintaining strong target-task gains and approaching the retention behavior of RL-based alternatives.

The main concerns were about framing and positioning rather than about whether the empirical effect is real. In particular, one reviewer questioned whether "on-policy SFT" is the right characterization, and others asked for clearer positioning of the PLD analysis, stronger baselines, more cost/scaling details, and sensitivity analyses. The rebuttal substantially strengthened the paper. Reviewers explicitly noted that the additional experiments and clarifications addressed their concerns about larger models, compute/storage overhead, checkpoint mixing, and the role of PLD, and one reviewer raised the score after the discussion. My reading of the discussion is that the remaining issues are best treated as revision requests for the final version, especially around precise terminology and more careful framing of the diagnostic claims. Overall, I find the paper to be a solid, technically convincing contribution and recommend accept.